# Research on Mobile Robot Path Planning Based on MSIAR-GWO Algorithm

**DOI:** 10.3390/s25030892

**Published:** 2025-02-01

**Authors:** Danfeng Chen, Junlang Liu, Tengyun Li, Jun He, Yong Chen, Wenbo Zhu

**Affiliations:** College of Mechanical Engineering and Automation, Foshan University, Foshan 528000, China; fosuliujl@163.com (J.L.); ltymxcz0000@163.com (T.L.); hejun_723@fosu.edu.cn (J.H.); cheny@fosu.edu.cn (Y.C.); zhuwenbo@fosu.edu.cn (W.Z.)

**Keywords:** path planning, gray wolf optimization algorithm, reinforcement learning, parameter selection, detour foraging

## Abstract

Path planning is of great research significance as it is key to affecting the efficiency and safety of mobile robot autonomous navigation task execution. The traditional gray wolf optimization algorithm is widely used in the field of path planning due to its simple structure, few parameters, and easy implementation, but the algorithm still suffers from the disadvantages of slow convergence, ease of falling into the local optimum, and difficulty in effectively balancing exploration and exploitation in practical applications. For this reason, this paper proposes a multi-strategy improved gray wolf optimization algorithm (MSIAR-GWO) based on reinforcement learning. First, a nonlinear convergence factor is introduced, and intelligent parameter configuration is performed based on reinforcement learning to solve the problem of high randomness and over-reliance on empirical values in the parameter selection process to more effectively coordinate the balance between local and global search capabilities. Secondly, an adaptive position-update strategy based on detour foraging and dynamic weights is introduced to adjust the weights according to changes in the adaptability of the leadership roles, increasing the guiding role of the dominant individual and accelerating the overall convergence speed of the algorithm. Furthermore, an artificial rabbit optimization algorithm bypass foraging strategy, by adding Brownian motion and Levy flight perturbation, improves the convergence accuracy and global optimization-seeking ability of the algorithm when dealing with complex problems. Finally, the elimination and relocation strategy based on stochastic center-of-gravity dynamic reverse learning is introduced for the inferior individuals in the population, which effectively maintains the diversity of the population and improves the convergence speed of the algorithm while avoiding falling into the local optimal solution effectively. In order to verify the effectiveness of the MSIAR-GWO algorithm, it is compared with a variety of commonly used swarm intelligence optimization algorithms in benchmark test functions and raster maps of different complexities in comparison experiments, and the results show that the MSIAR-GWO shows excellent stability, higher solution accuracy, and faster convergence speed in the majority of the benchmark-test-function solving. In the path planning experiments, the MSIAR-GWO algorithm is able to plan shorter and smoother paths, which further proves that the algorithm has excellent optimization-seeking ability and robustness.

## 1. Introduction

With the continuous development of computer and sensor technologies, mobile robots have been widely used in the fields of logistics and warehousing [1], agriculture [2], medical services [3], surveillance [4], and mining [5]. Autonomous navigation of mobile robots mainly involves four core tasks, perception, localization, path planning, and motion control [6], among which path planning, as an important part of safe and efficient driving, is one of the most critical technologies in robot navigation. Efficient path planning algorithms can not only dramatically improve the efficiency of robot picking and handling in production lines but also reduce robot wear and tear and capital investment. Path planning can be regarded as an optimization task, with path length, energy consumption, time, and smoothness as the optimal path selection metrics [7], aiming to find a collision-free optimal path that allows the robot to efficiently transfer from the initial state to the target area in a given environment under multiple constraints. Commonly used path planning methods for mobile robots are mainly categorized into classical path planning algorithms and meta-heuristics [8]. Classical path planning algorithms mainly include the A* algorithm [9], Dijkstra’s algorithm [10], the probabilistic roadmap method [11], the fast search random tree algorithm [12], and the dynamic programming method [13]. These methods have high computational efficiency and good interpretability but usually can only cope with simpler scenarios, and the computational difficulty in complex environments increases exponentially with the increase in the complexity of the environment [14], resulting in high computational costs and low success rates, making it difficult to meet the needs of practical applications.

Meta-heuristic algorithms, by combining heuristics and stochastic search strategies, have powerful global search capability and high versatility, perform well in solving complex optimization problems, and have a wide range of application prospects. Meta-heuristic algorithms can be broadly classified into physics-based algorithms, evolution-based algorithms, and group intelligence optimization algorithms according to different principles and mechanisms, such as simulating physical phenomena in nature, biological evolutionary processes, and group intelligence behaviors. Obstacles in the path planning problem divide the complex search space into multiple regions, and there may be multiple local optimal solutions in these regions, making the problem present nonconvex characteristics [15]. In recent years, path planning as a nondeterministic polynomial time problem (NP) [16] has become a research hotspot, with researchers attempting to solve it using swarm intelligent optimization algorithms, such as the particle swarm algorithm [17], ant colony algorithm [18], sparrow search algorithm [19], etc. Lin Xu et al. [20] proposed a new particle swarm algorithm based on a quadratic Bezier transition curve and an optimized particle swarm algorithm. Fengcai Huo et al. [21] proposed an improved ant colony algorithm based on corner constraints and an improved b-spline curve-smoothing algorithm considering minimum-turning-radius constraints, which can plan a balanced path length and turning-frequency path with a faster convergence speed. Yao Cheng et al. [22] proposed an improved sparrow search algorithm combined with a chaos optimization algorithm regardless of whether it is an optimization algorithm or an optimized particle swarm algorithm. The improved sparrow search algorithm, which shows obvious improvement in both global and local searching abilities, enables unmanned underwater vehicles to find more reasonable and safer paths in a 3D environment.

The gray wolf optimizer (GWO), as one of the classical group intelligence optimization algorithms, mainly simulates the social hierarchy and hunting behavior of gray wolves; is popular due to the advantages of having few adjustable parameters, high efficiency, and a simple and easy-to-implement structure; and is widely used in image processing [23], electric power scheduling [24], feature selection [25], and shop scheduling [26]. Although the gray wolf optimization algorithm performs well in many optimization problems, it has some limitations. In solving some complex problems, the gray wolf optimization algorithm still faces problems such as low convergence accuracy, an imbalance between global exploration and local exploitation, and a tendency to fall into local optimization. In order to enhance the performance of the gray wolf optimization algorithm, many scholars have improved the algorithm from different aspects and proposed various GWO variants to overcome the above shortcomings, which can be roughly classified into the following types. The first one is to adjust the position-update equation as well as to design the mutation strategy. For example, Lili Liu et al. [27] proposed an adaptive position-update strategy that combines the Levy flight and the golden sine, which improves the algorithm’s solution accuracy and global search capability by combining the long-distance jumps of Levy flights for the global search in the search space with the golden sine to guide to the more promising regions. Yijie Zhang et al. [28] introduced a dynamic logarithmic spiral that decreases nonlinearly with the number of iterations to overcome the shortcomings of the traditional gray wolf optimization algorithm, which approaches the leader along a straight line and tends to ignore the information on the path; they proposed a new position-updating strategy using globally optimal and randomly generated positions as learning samples. This can dynamically control the influence of the learning samples in order to increase the diversity of the population and keep the algorithm from converging too early. The second one is combined with other optimization algorithms to make full use of the advantages of the algorithm. For example, Ishaq Ahmad et al. [29] enhanced the exploitation capability by incorporating onlooker and scout bee operations in the artificial bee colony algorithm into the position change phase of the gray wolves, thus improving the local convergence efficiency. Binbin Tu et al. [30] endowed the position-update process of the gray wolves with a hawk-like flight capability and a broad field of view by combining these with the Harris Hawk optimization algorithm to improve the global search ability, further accelerating the convergence speed of the population. The third is to adjust and optimize the control parameter. In the traditional gray wolf optimization algorithm, the control parameter *A* is a research hotspot as an important influencing factor to balance the global search and local exploitation ability. The control parameter *A* depends on the convergence factor *a*. A common improvement method is to adjust the traditional linear convergence factor, which decreases linearly with the number of iterations, to a nonlinear function to improve the exploration. For example, Di Zhao et al. [31] improved the linear convergence factor into an exponential decay function and obtained the parameters controlling the search interval and the curvature of the function by testing. The improved nonlinear convergence factor improves the global search ability in the early stage and the local search ability in the late stage, which balances the two kinds of searching abilities of the algorithm and improves the efficiency of the algorithm as much as possible. H. Nasiri Soloklo, N. Bigdeli [32] used a sigmoid function instead of a linear function and adjusted the scaling factor and curvature to achieve a variety of search ranges and rates of change in different parts of the domain; by adjusting the parameters, the exploration and development stages of the algorithm can be controlled more efficiently.

Upon analysis, it is found that the traditional gray wolf optimization algorithm (GWO) has limited performance and is insufficient for solving more complex optimization problems. Although many improved gray wolf optimization algorithms have significantly improved performance and generated high-quality solutions in practical application scenarios, these algorithms all contain nonlinear convergence factors with adjustable parameters for controlling the search space and function curvature, and the adjustable parameters vary with different search mechanisms. The selection of these parameters can only be gradually adjusted through multiple sets of experiments or selected based on empirical values, which is difficult to quantitatively portray and analyze. When the number of experimental samples is not large enough, it is easy to miss the optimal parameter combination, which in turn affects the generation of optimal planning solutions. Based on the above analysis, this paper proposes a multi-strategy improved gray wolf optimization algorithm (MSIAR-GWO) based on reinforcement learning and verifies the effectiveness of this algorithm in robot path planning through simulation experiments. The main innovations of this paper are reflected in the following points:(1)A new nonlinear convergence factor is proposed, and adjustable parameters are intelligently selected through reinforcement learning to adapt to specific variants of the gray wolf optimization algorithm based on the improvement of different strategies, which enables the optimization process to find a balance between exploration and exploitation. Intelligent configuration of adjustable parameters through reinforcement learning can reduce human intervention and improve the robustness and adaptability of the algorithm.(2)A new adaptive position-updating strategy based on detour foraging and dynamic weights is proposed. Dynamic weights can be dynamically assigned in the iterative process according to the change in the adaptation value characterizing the size of the role played by different types of gray wolves in the leadership, increasing the weights of the more optimal individuals and accelerating the convergence speed of the algorithm as a whole. At the same time, an adaptive position-update mechanism is added to ensure that the diversity of the wolf pack can still be maintained when the wolf pack gathers to the leadership in the late iterations. Since the position-update mechanism of the traditional gray wolf optimization algorithm mainly relies on the guidance of the leader wolf, the whole optimization process lacks information sharing and collaboration among individuals, which to some extent affects the algorithm’s search diversity and global optimization ability. For this reason, we further add the detour foraging mechanism of the artificial rabbit optimization algorithm to the position-updating strategy of the gray wolf optimization algorithm, and we add Levy flight strategy or a Brownian motion strategy to the detour foraging mechanism of the artificial rabbit algorithm according to the energy factor. This enhances the information sharing of individuals in the population and then enriches the path diversity among individuals so that the algorithm has a significant advantage in solving complex optimization problems.(3)We introduce an elimination and relocation strategy based on stochastic center-of-gravity dynamic reverse learning for the inferior individuals in the population to improve the search range of wolf individuals and keep the algorithm from falling into local optimum.

## 2. Basic Theory

### 2.1. Overview of the Gray Wolf Optimization Algorithm

The gray wolf optimization algorithm is a group intelligence optimization algorithm based on the hunting behavior of gray wolves proposed by Mirjalili et al. [33] in 2014. The gray wolf is a canid with an obvious group-living tendency and has a strict social hierarchy. The gray wolf population can be divided into four tiers according to the social status from high to low, α wolf, β wolf, δ wolf, and ω wolf, which are in the shape of a pyramid, as shown in Figure 1. The α wolf occupies the highest status in the wolf pack and is responsible for hunting, territorial defense, and decision-making within the group, and the β wolf, inhabiting the second tier of the pack, is subordinate to the α wolf and assists the α wolf in decision-making and managing the other gray wolves. The δ wolf, in the third tier, is subordinate to the α wolf and β wolf and further supplements their guiding roles, and it is mainly responsible for scouting and sentry duty. In contrast, ω wolves are at the bottom of the social hierarchy and are responsible for maintaining the balance of relationships within the population. ω wolves follow the guidance of the α wolf, β wolf, and δ wolf and explore, updating their position. The hunting process of gray wolves generally consists of three steps: encircling the prey, hunting, and attacking the prey.

#### 2.1.1. Surround the Prey

When the gray wolf searches for prey, it needs to calculate the current distance between itself and the prey; it gradually approaches the prey and encircles it, and the mathematical model of the gray wolf encircling the prey can be expressed as(1)$$D=\left|C\times X_{P}\left(t\right)-X\left(t\right)\right|$$(2)$$X\left(t+1\right)=X_{P}\left(t\right)-A\times D$$

*D* is the distance between the individual and the prey, *t* is the current number of iterations, $X_{P}\left(t\right)$ and X(t) are the current positions of the prey and the corresponding $X_{P}\left(t\right)$ and X(t) are the current positions of the prey and the corresponding gray wolf at iteration *t*. gray wolf at iteration *t*, and *A* and *C* are the coefficient vectors computed from Equation (Equation 3) and Equation (Equation 4), respectively.(3)$$A=2\times r_{1}\times a-a$$(4)$$C=2\cdot r_{2}$$(5)$$a=2-2\times \frac{t}{t_{\mathrm{MaxIter}\;}}$$

Both $r_{1}$ and $r_{2}$ are random numbers between [0,1], and $t_{\mathrm{MaxIter}\;}$ is the total number of iterations.

#### 2.1.2. Hunting

Gray wolves are unable to determine the precise location of their prey, but they have the ability to identify the location of potential prey, and since α, β, and δ are considered to have a greater probability of identifying prey, other individual gray wolves update their own positions based on these wolves’ positions, as shown in Figure 2. During the hunting process, the basic position-updating method of the gray wolf is defined as(6)$$\begin{array}{cc} D_{\alpha } & =\left|C_{1}\times X_{\alpha }-X\right|\\ D_{\beta } & =\left|C_{2}\times X_{\beta }-X\right|\\ D_{\delta } & =\left|C_{3}\times X_{\delta }-X\right| \end{array}$$(7)$$\begin{array}{cc} \; & X_{1}=X_{\alpha }-A_{1}\times D_{\alpha }\\ \; & X_{2}=X_{\beta }-A_{2}\times D_{\beta }\\ \; & X_{3}=X_{\delta }-A_{3}\times D_{\delta } \end{array}$$(8)$$X\left(t+1\right)=\frac{X_{1}+X_{2}+X_{3}}{3}$$
where $D_{\alpha }$, $D_{\beta }$, and $D_{\delta }$ are the distances between the current gray wolf and α, β, and δ, respectively. $X_{\alpha }$, $X_{\beta }$, and $X_{\delta }$ represent the positions of α, β, and δ, respectively. $C_{1}$, $C_{2}$, and $C_{3}$ are random vectors, and *X* is the current position of the gray wolf. Equation (Equation 8) represents the position-update formula for an individual gray wolf.

#### 2.1.3. Attacking Prey

In the mathematical model of attacking prey, *A* is a random number varying between [−*a*, *a*], and the range of fluctuation of *A* is reduced by controlling *a* to decrease linearly during the iteration process. When *A* is in the interval [−1, 1], then the letting agent’s next momentary position can be anywhere between the current gray wolf and its prey, and when |A|<1, this forces an attack on the prey, prompting the wolves to perform a local search.

### 2.2. Fundamentals of the Detour Foraging Strategy of the Artificial Rabbit Optimization Algorithm

Artificial rabbit optimization (ARO) is a novel meta-heuristic optimization algorithm proposed by Liying Wang et al. [34] in 2022 inspired by the survival strategies of rabbits in nature. It uses the foraging and hiding strategies of real rabbits and switches between the two strategies through energy contraction. Characterized by strong search ability, fast convergence, and adaptability, detour foraging is a practice where rabbits do not forage in their own area but always randomly detour to grass foraging near other rabbits’ nests in order to prevent their nests from being detected by predators. This detour foraging strategy is actually likely to disturb the area around the food source in order to obtain enough physical objects, and its mathematical model is represented as follows:(9)$$\begin{array}{c} p_{i}\left(t+1\right)=y_{j}\left(t\right)+R\cdot \left(y_{i}\left(t\right)-y_{j}\left(t\right)\right)+\mathrm{round}\left(0.5\cdot \left(0.05+r_{3}\right)\right)\cdot n_{1}\\ i,j=1,\ldots ,n\;\mathrm{and}\;j\neq i \end{array}$$(10)R=S·c(11)$$S=\left(e-e^{{\left(\frac{t-1}{t_{\mathrm{MaxIter}\;}}\right)}^{2}}\right)\times \sin\left(2\pi r_{4}\right)$$(12)$$c\left(k\right)=\left\{\begin{array}{cc} 1 & \;\mathrm{if}\;k==\varphi (l)\\ 0 & \;\mathrm{else}\; \end{array}k=1,\ldots ,d\;\mathrm{and}\;l=1,\ldots ,\left\lceil r_{5}\cdot d\right\rceil \right.$$(13)$$n_{1}∼N\left(0,1\right)$$(14)φ=randperm(d)(15)$$E\left(t\right)=4\cdot \left(1-\frac{t}{t_{\mathrm{MaxIter}\;}}\right)\cdot \ln\frac{1}{r}$$
where $p_{i}\left(t+1\right)$ is the candidate solution of the *i*th rabbit at the t+1st iteration, and $y_{j}\left(t\right)$ is the current position of the *j*th rabbit at the *t*th iteration; $n_{1}$ is a random numbers obeying the normal distribution; round denotes rounding; *R* is the running operator, which is used to simulate the running characteristics of the rabbits; *S* denotes the step length of the running; *d* is the dimensionality of the problem; *n* is the size of the rabbit population; $t_{\mathrm{MaxIter}\;}$ is the maximum number of iterations; $r_{3}$, $r_{4}$, and $r_{5}$ are random numbers between [0,1]; ⌈·⌉ is the upward rounding function; φ is a random integer between 1 and *d*; and *E* is the energy factor, which decreases with time.

## 3. Multi-Strategy Improved Gray Wolf Optimization Algorithm Based on Reinforcement Learning (MSIAR-GWO)

### 3.1. Nonlinear Convergence Factors for Optimization Based on Reinforcement Learning Algorithms

Group intelligence optimization algorithms generally have the problem of balancing global search capability and local search capability, and the gray wolf optimization algorithm is no exception. From Equations (Equation 6) and (Equation 7), it can be seen that when |A|⩾ 1, the wolves are far away from the prey and search globally in the whole search space, and a stronger global search performance can effectively help the algorithm to avoid falling into the local optimal solution; when |A|<1, the wolves make use of the collected information to conduct an accurate search in the local area and gradually approach the prey, and the strong local development performance can improve the algorithm’s solving accuracy and accelerate the convergence of the algorithm. The strong local exploitation performance can improve the algorithm’s solution accuracy and accelerate the convergence speed of the algorithm. Equation (Equation 3) shows that the size of |A| is determined by the convergence factor *a*. The convergence factor *a* of the traditional gray wolf optimization algorithm decreases linearly from 2 to 0 with iterations, with half of the iterations for exploration and half for exploitation, as shown in Figure 3. However, this linear variation does not reflect the actual search process and is insufficient to adapt to the needs of complex optimization problems. Therefore, a new nonlinear convergence factor is introduced to reasonably characterize the actual optimization process, and the specific mathematical model expression of the improved convergence factor is(16)$$a=a_{\mathrm{final}\;}+\left(a_{\mathrm{initial}}-a_{\mathrm{final}}\right)\cdot {\left(1-{\left(\frac{t}{t_{\mathrm{MaxIter}\;}}\right)}^{\lambda_{1}}\right)}^{\lambda_{2}}$$
where $a_{\mathrm{initial}\;}$ and $a_{\mathrm{final}\;}$ are the initial and termination values of the convergence factor *a*, respectively, *t* is the current number of iterations, $t_{\mathrm{MaxIter}\;}$ is the maximum number of iterations, and $\lambda_{1}$ and $\lambda_{2}$ are the nonlinear adjustment coefficients.

The nonlinear convergence factors proposed in the literature [31,32,35,36] all have nonlinear tuning parameters, which can only be selected by gradual adjustment through multiple sets of experiments or qualitatively analyzed with extreme reliance on empirical values, and they are difficult to accurately represent quantitatively. If the number of parameter samples used for comparison experiments is not large enough, it is easy to miss the best parameter combination. Reinforcement learning is a machine learning method that incorporates five elements: environment, intelligences, states, actions, and rewards. By modeling the human characteristic of learning from experience, the intelligent body is able to obtain feedback rewards from the environment after executing an action in a specific state. It also guides the intelligent body to learn how to take actions to maximize the cumulative rewards by repeated trial and error and adjustments in its interaction with the environment and finally achieve the optimal strategy. Therefore, a new strategy for determining the nonlinear adjustment parameters $\lambda_{1}$ and $\lambda_{2}$ in the nonlinear convergence factor of Equation (Equation 16) is proposed. Since the algorithm’s optimality-seeking performance is more significantly affected by parameter variations when solving the single-peak function Step and the multi-peak function Penalized1, they are used as test functions and are used to intelligently select the nonlinear regulation parameters in the gray wolf optimization algorithm without relying on the past empirical values by utilizing the value-based Q-learning algorithm. Q-learning is used to determine the nonlinear regulation parameters in the nonlinear convergence factor of Equation (Equation 16) by constantly updating the Q-values of state–action pairs in the Q-value table to gradually approximate the optimal policy, thus guiding the agent to choose the optimal action in different states. The Q-values stored in the Q-table can be updated according to the rewards through the state–action value function with the following formula:(17)$$Q_{t+1}\left(s_{t},a_{t}\right)=Q\left(s_{t},a_{t}\right)+\tau \left[r_{t+1}+\gamma \max_{a}Q\left(s_{t+1},a\right)-Q\left(s_{t},a_{t}\right)\right]$$
where the variables $s_{t}$ and $s_{t+1}$ denote the current state and the next state, respectively, $a_{t}$ is the current action, γ is the discount factor, τ is the learning rate, $r_{t+1}$ is the reward obtained when executing the action $a_{t}$ in the state $S_{t}$, and $Q\left(s_{t+1},a\right)$ is the estimated *Q* value when executing the action *a* in the state $s_{t+1}$.

Considering the gray wolf population as an intelligent body, the ultimate goal is to output the optimal parameter combinations $\lambda_{1}$ and $\lambda_{2}$. The specific parameter optimization process for the state set, action set, and reward method is described as follows:

(1) State: construct the state vector $W_{i}\left(W={\left[\lambda_{1,i},\lambda_{2,i}\right]}^{T}\right)$ consisting of nonlinear regulation parameters $\lambda_{1}$ and $\lambda_{2}$, where *i* denotes the number of iterations; set the parameter’s optimization space as [1, 10]; and finally, randomly assign the initial parameter vector $W_{0}$ in the search range.

(2) Action: The action of reinforcement learning is to determine the changes in the state vector $W_{i}$. The changes in the state vectors $\lambda_{1}$ and $\lambda_{2}$ during the reinforcement learning process can be divided into three types: increasing, unchanged, and decreasing. The change step size for each iteration is 1, and it can be divided into nine types of actions based on the changes.

(3) Reward: Through comprehensive evaluation of the algorithm’s optimization accuracy and its stability performance, taking the optimal value and standard deviation as two indicators of the adaptation value, we then measure the state vector as good or bad. If the state vectors between two neighboring generations are different after the agent executes the action, and remembering that the smaller the value of the fitness function is, the higher the algorithm’s optimization performance is, then it will be positively rewarded. If the state vectors between two neighboring generations are the same after the agent executes the action and if the historical optimal fitness value of the state is optimal in the historical data set, it is positively rewarded.

### 3.2. Adaptive Position-Update Strategy Based on Detour Foraging and Dynamic Weighting

In the basic GWO, since α, β, and δ wolves are closest to the prey, their positions are used to estimate the approximate location of the prey. However, if they all fall into the local optimum and the ω wolves in the population still converge to the positions of these wolves, it means that the leadership of the wolf pack misjudges the position of the prey, and the population cannot explore enough in the search space and converges prematurely, leading to difficulties in finding a better solution. In this regard, in order to enhance the ability of the algorithm to jump out of the local optimum and to maintain the population diversity, an adaptive position-update formula is proposed, as shown in Equation (Equation 18), which incorporates a perturbation along with the consideration of randomly selecting another individual in the population with a better fitness value than the current fitness value as well as the information of the individual with the optimal fitness value to guide the search of the candidate individuals.(18)$$X\left(t+1\right)=\frac{X1+X2+X3}{3}+l_{1}\cdot \left(\mu_{1}\cdot \rho_{1}\cdot \left(X^{\prime }-X\left(t\right)\right)+\mu_{2}\cdot \left(1-\rho_{1}\right)\cdot \left(X_{\alpha }-X\left(t\right)\right)\right)$$(19)$$\rho_{1}=1-\frac{t}{T}$$
where $\rho_{1}$ is a weight reflecting the influence of $X^{\prime }$ and $X_{\alpha }$ at different iteration moments. The $\rho_{1}$ weight indicates that in the initial stage of the search, it is more influenced by the information of the randomly selected excellent individuals to make the position-update equation sufficiently stochastic, utilizing more useful information for stochastic exploration. In the later stages of the search, more attention is paid to the role of the optimal individuals to perform a localized and finer search to generate more promising candidate individuals. Here, $l_{1}$ is a constant controlling the size of randomness, which is taken as 0.3 in this paper. $\mu_{1}$ and $\mu_{2}$ are random numbers between (0, 1), and *T* is the maximum number of iterations.

In the traditional GWO, the average of the positions of α, β, and δ wolves in the iterative position-updating formula are used to update the position of the whole population, and the three leadership individuals have the same guidance for the group; however, the gray wolf optimization algorithm itself is based on the algorithm of a social hierarchical relationship, so the equivalent guidance makes the algorithm converge slowly. In this regard, the present invention proposes a dynamic proportional weight strategy based on the value of the fitness to highlight the importance of the relationship between the α, β, and δ wolves; using the degree of importance between the three, the proportional weight calculation formula based on the adaptability value is shown in Equations (Equation 21)–(Equation 23). Improving the position-update formula can make the algorithm converge to the optimal solution faster than Equation (Equation 20).(20)$$X\left(t+1\right)=\frac{W_{\alpha }\cdot X1+W_{\beta }\cdot X2+W_{\delta }\cdot X3}{W_{\alpha }+W_{\beta }+W_{\delta }}+l_{1}\cdot \left(\mu_{1}\cdot \rho_{1}\cdot \left(X^{\prime }-X\left(t\right)\right)+\mu_{2}\cdot \left(1-\rho_{1}\right)\cdot \left(X_{\alpha }-X\left(t\right)\right)\right)$$(21)$$W_{\alpha }=\frac{f_{\alpha }+f_{\beta }+f_{\delta }}{f_{\alpha }}$$(22)$$W_{\beta }=\frac{f_{\alpha }+f_{\beta }+f_{\delta }}{f_{\beta }}$$(23)$$W_{\delta }=\frac{f_{\alpha }+f_{\beta }+f_{\delta }}{f_{\delta }}$$
where $f_{\alpha }$, $f_{\beta }$, $f_{\delta }$ denote the adaptation values of α, β, δ wolves.

Gray wolf optimization algorithms are prone to fail to explore the search space efficiently in dealing with complex problems such as multi-peak optimization, resulting in falling into local optima and converging prematurely. In the basic GWO algorithm, the individuals in the wolf pack as followers are only guided by the information of the three leader wolves, and there is a lack of information sharing among the individuals of the pack, whereas in the detour foraging behavior of the ARO algorithm, each searching individual ignores the food in its vicinity and tends to update its position to the other searching individuals randomly selected in the population, and the individuals other than the leaders are given the opportunity to guide the rabbit pack, which enhances the pack’s exploration ability. To this end, a novel hybrid algorithm based on the GWO algorithm and the ARO algorithm is proposed to avoid the stagnation of the GWO algorithm in the exploitation phase, introducing the detour foraging behavior of the ARO algorithm at *a* < 1. The gray wolves enhance the exploration capability of the candidate solutions by learning the detour foraging behavior of the rabbits in the MSIAR-GWO algorithm so that the group is equipped with the ability to share information among individuals while the wolves maintain the original hunting strategy to retain its exploitation capability. In order to trade off the performance of exploration and exploitation and improve the accuracy of optimization, Levy flight and Brownian motion strategies are also introduced to the detour foraging process of the artificial rabbit optimization algorithm. Inspired by the energy contraction process of the artificial rabbit optimization algorithm, the conversion process from exploration to exploitation is simulated by the energy factor, with the energy factor E=1 as the cut-off point. When E>1, the Levy flight strategy is used for global search, and since the Levy flight is a stochastic wandering strategy with jumping characteristics, its step length obeys the heavy-tailed distribution; the distribution of the step lengths is uneven, with most of the steps being shorter but with an occasional large jump, as shown in Figure 4a. This is conducive to crossing to other locations with a large probability in the early iterations, so that the gray wolf individuals are widely distributed in the search space in order to improve the global optimality-finding ability. When *E*⩽ 1, the Brownian motion strategy is used for local detection, because the Brownian motion is a kind of description of the irregular random motion of the particles in fluid over short distances, with uniform distribution of the direction and distance of the movement of each step, and there are no obvious long-distance jumps, as shown in Figure 4b. This facilitates a finer and more continuous search of the solution space when the optimal solution is approached in later iterations, which helps the algorithm to find a better solution in the local region to improve the accuracy of the algorithm. The improved formula is shown in Equation (Equation 24).(24)$$Y\left(t+1\right)=\left\{\begin{array}{c} X^{\prime }\cdot \mathrm{Levy}\left(d\right)+R\cdot \left(X\left(t\right)-X^{\prime }\right)+\mathrm{round}\left(0.5\cdot \left(0.05+r_{1}\right)\right)\cdot n_{2},E>1\\ X_{\alpha }+l_{2}\cdot R\cdot B_{t}\cdot \left(X\left(t\right)-X_{\alpha }\cdot B_{t}\right)+\mathrm{round}\left(0.5\cdot \left(0.05+r_{2}\right)\right)\cdot n_{3},E\leq 1 \end{array}\right.$$
where Y(t+1) is the position of the candidate solution for the detour foraging behavior at the t+1st iteration; $X^{\prime }$ is the position of the random individual in the population with a higher fitness value than the current position; X(t) is the current position at the *t*th iteration; *R* is the running operator represented by Equation (Equation 10); $l_{2}$ is the constant controlling the magnitude of the effect of the Brownian motion, which is set to 10 in this paper; *E* is the energy factor in Equation (Equation 15); $n_{2}$ and $n_{3}$ are random numbers obeying the standard normal distribution; and $X_{\alpha }$ is the position of the individual with the highest fitness value of the population. Levy(d) is the Levy flight distribution function, where *d* is the dimension of the problem, and its calculation formula is as follows:(25)$$\mathrm{Levy}\left(d\right)=s\cdot \frac{u\cdot \sigma }{{\left|v\right|}^{\frac{1}{\eta }}}$$
where *s* is a fixed constant 0.005; β is a correlation parameter whose value is set to 1.5 in this paper; *u* and *v* are random numbers in the interval [0,1]; and σ is calculated as follows:(26)$$\sigma ={\left(\frac{\Gamma \left(1+\eta \right)\cdot \sin\left(\frac{\pi \eta }{2}\right)}{\Gamma \left(\frac{1+\eta }{2}\right)\cdot \eta \cdot 2^{\left(\frac{\eta -1}{2}\right)}}\right)}^{\frac{1}{\eta }}$$

Γ denotes the standard Gamma function, and the value of η is 1.5. $B_{t}$ is the random wandering coefficient of Brownian motion, which is essentially a random value that obeys the standard normal distribution of points, and it can be obtained from Equation (Equation 27).(27)$$B_{t}=\frac{1}{\sqrt{2\pi }}\exp\left(-\frac{x^{2}}{2}\right)$$

Although the detour foraging behavior improved by Levy flight and Brownian motion strategies can be obtained to achieve the position update by the guidance of the position information of the other gray wolves in the population except the leader, there is no guarantee that the fitness of the new solution obtained by Equation (Equation 24) is better than that of the solution derived from the position-update formula of Equation (Equation 20), and therefore, a greedy mechanism is used to compare the fitness of these two solutions in order to retain the solution with the better fitness.(28)$$X\left(t+1\right)=\left\{\begin{array}{c} X(t+1),f(X(t+1))<f(Y(t+1))\\ Y(t+1),f(Y(t+1))\leq f(X(t+1)) \end{array}\right.$$

### 3.3. Stochastic Center-of-Gravity-Based Dynamic Reverse Learning for Elimination and Relocation Strategy

In the traditional GWO algorithm, the inferior individuals involved in the process of updating the position and searching for the best solution contribute less, but the ineffective individuals still occupy computational resources during the evolution process, which reduces the overall search efficiency. The limited exploration capability of the inferior individuals leads to the inability to effectively explore the entire search space. Their existence makes the search path too concentrated in certain inefficient regions, restricting the ability of the algorithm to jump out of the local optimum, making it difficult for the population to maintain a stable and efficient search path, and affecting the search for the global optimal solution. Therefore, the present invention eliminates and re-updates the localization of the inferior individuals ranked in the bottom 10% of the fitness value through stochastic gravity dynamic reverse learning, which can effectively alleviate these adverse effects, improve the search range of wolf individuals, keep the algorithm from falling into the local optimum, improve the diversity of the population, and accelerate the convergence process. Tizhoosh [37] showed that the inverse solution of elite reverse learning has a 50% probability of being closer to the global optimal solution. Stochastic center-of-gravity dynamic reverse learning takes into account the concepts of center of gravity and adversarial learning, considering the importance of elite individual guidance, randomly selecting elite individuals in the population according to the dynamic changes in the current iteration process to compute a center-of-gravity point, and generating reverse individuals based on it, which enhances the robustness and flexibility of the algorithm through the introduction of stochastic elements and dynamics.

We randomly generate an integer *n∈[1,N]*, where *N* is the population size, select the top *n* individuals $X_{1},X_{2}\ldots X_{n}$ with the best adaptation in the current wolf population, and calculate the center of gravity of these *n* individuals as shown in Equation (Equation 29).(29)$$M=\frac{\sum_{i=1}^{n}X_{i}}{n}$$

The inverse solution is calculated based on the desired center of gravity as shown in Equation (Equation 30).(30)$$X_{i}^{*}=2\times M-X_{i},i=1,2,\ldots N$$

A greedy strategy is utilized to select the optimal-solution individual among the current solution and its inverse solution as the new-generation individual.(31)$$X_{i}=\left\{\begin{array}{c} X_{i},f\left(X_{i}\right)<f\left(X_{i}^{*}\right)\\ X_{i}^{*},f\left(X_{i}^{*}\right)<f\left(X_{i}\right) \end{array}\right.$$

### 3.4. Flowchart of MSIAR-GWO Algorithm

Based on the above discussion, this paper proposes a multi-strategy improved gray wolf optimization algorithm based on reinforcement learning optimization (MSIAR-GWO) by combining a nonlinear convergence factor based on reinforcement learning optimization, an adaptive position-updating strategy based on detour foraging and dynamic weights, and an elimination and relocation strategy based on stochastic center-of-gravity dynamic inverse learning, and the flowchart of MSIAR-GWO is shown in Figure 5. It effectively improves the shortcomings of the GWO algorithm such as slow convergence speed, ease of falling into the local optimum, and low convergence accuracy when dealing with high-dimensional complex problems.

## 4. Experimental Verification

### 4.1. Benchmarking Function Optimization and Result Analysis

In order to examine the optimization-seeking performance of the MSIAR-GWO algorithm proposed in this paper, it is compared with other common swarm intelligence optimization algorithms such as the original gray wolf optimization algorithm (GWO), the artificial rabbit optimization algorithm (ARO), the dung beetle optimization algorithm (DBO), and the whale optimization algorithm (WOA), as well as with a series of other gray wolf optimization algorithms based on the gray wolf optimization algorithm such as the IGWO, the AGWO, the RSMGWO, and other improved algorithms. In addition, simulation experiments are conducted on 18 standard test functions, among which, F1–F7 shown in Table 1 are single-peak benchmark test functions for examining the convergence speed and accuracy of the algorithms; the multi-peak benchmark test functions F8–F13 in Table 2 and the fixed-dimension multi-peak benchmark test functions F14–F18 in Table 3 are used to examine the exploration ability of the algorithms and their ability to avoid falling into the local optimum. In order to ensure the fairness of the simulation experiments, the maximum number of iterations of all algorithms is set to 500, and the number of populations is set to 30. In order to eliminate the influence of randomness, all the experiments of the above algorithms are executed separately 20 times; and the optimal value, average value, worst value, and standard deviation of each algorithm are calculated; the final evaluation of the searching accuracy is made by the average value; and the standard deviation is evaluated by the stability performance. All tests were performed using the following hardware and software environments: 12th Gen Intel(R) Core(TM) i5-12600KF CPU, 370 GHz, 10 cores, 16 logical processors; 32 GB RAM; NVIDIA GeForce RTX 4080 Laptop GPU; Microsoft Windows 11 operating system; Matlab; and NVIDIA GeForce RTX 4080 Laptop GPU; Windows 11 operating system; Matlab 2021a programming software.

F1–F7 as single-peak test functions have only one global optimal solution, so they can be used to evaluate the performance of algorithm development. From Table 4, it can be concluded that the MSIAR-GWO algorithm proposed in this paper and the RSMGWO algorithm show similar convergence accuracy in F1–F4, and both of them can obtain the theoretical optimal solution, but from Figure 6a–d, it can be seen that the MSIAR-GWO can find the minimum value quickly in less than 150 iterations, which shows a high search efficiency, and the convergence speed of the optimization search process is faster than that of the RSMGWO, while the convergence results of GWO and IGWO are still significantly different from the theoretical optimum. By comparing the experimental results in Table 4, it can be seen that MSIAR-GWO is always in first place in the optimization performance in F5–F7 compared with the other comparison algorithms in terms of the optimal value, average value, worst value, and standard deviation, showing better optimization ability and excellent stability. As shown in Figure 6e–g, in the first 50 iterations, all the algorithms have faster convergence speeds at the beginning of the iterations and have similar convergence curves, but after 50 iterations, although all the algorithms’ convergence speeds have slowed down significantly, MSIAR-GWO still has faster convergence speeds than the other algorithms to continue to search for a better solution in the search space, and the exploration of the convergence curves gradually deepens. However, MSIAR-GWO still has a faster convergence rate than the other algorithms to continue searching for better solutions in the search space, and the exploration of the convergence curve gradually deepens. This is thanks to the intelligent selection of adjustable parameters in the proposed nonlinear convergence factor through reinforcement learning in this paper, which can well measure the exploration and exploitation process of the population and successfully maintains the balance between the diversity and convergence of the population. The dynamic incorporation of Levy flight and Brownian motion strategies for the detour foraging behavior allows the algorithm to explore the search space more fully by incorporating more stochastic operations, so that the algorithm can achieve more precise convergence accuracy. Moreover, the adaptive position-update formula based on dynamic weights can well highlight the importance of different dominant individuals according to the hierarchical relationship, and the improved algorithm shows rapid convergence speed.

F8–F13 as multi-peak test functions have many local optimal solutions in the search space and thus can be a good test of an algorithm’s ability to solve complex problems. As shown in Table 5, Although MSIAR-GWO fails to achieve the best result in F8, its optimization result is only after RSMGWO and WOA; its optimization ability is in third place, and the algorithms AGWO, ARO, RSMGWO, and MSIAR-GWO can finally converge to the theoretical minimum in F9; ARO, DBO, WOA, AGWO, RSMGWO and MSIAR-GWO algorithms can also converge to the theoretical optimal value in F11; the convergence accuracy of ARO, DBO and MSIAR-GWO algorithms in F10 is the same. Although there are other algorithms in F9, F10, and F11 that have the same convergence accuracy as the MSIAR-GWO algorithm and finally converge to the same solution, it can be seen in Figure 7b–d that compared with the traditional GWO, which once it falls into the local optimum finds it difficult to escape, leading to premature convergence, the MSIAR-GWO finds the theoretically optimal solution in less than 50 iterations, and the convergence speed is much faster than other algorithms under the same conditions. In the benchmark test functions F12 and F13, our method MSIAR-GWO shows an absolute advantage over other classical optimization algorithms and other improved gray wolf optimization algorithms with all the statistical data. For the multi-peak test functions, it is demonstrated through simulation experiments that the MSIAR-GWO algorithm, by incorporating an adaptive position-updating strategy based on detour foraging and dynamic weights as well as a stochastic center-of-gravity dynamic reverse learning elimination and relocation strategy, increases the diversity of the population while guaranteeing the quality of the elite individuals in the process of population evolution, significantly reduces the risk of falling into a local optimum, and improves the search ability of the algorithm.

F14–F18 are fixed-dimension multi-peak benchmark functions whose peaks’ number and position remain stable throughout the search process, which helps to more accurately evaluate the performance of the algorithms in dealing with the stable multi-modal structure problem. MSIAR-GWO, ARO, DBO, and IGWO all eventually converge to the same minimum in F14, but the average value of IGWO is slightly better than that of the MSIAR-GWO algorithm, and MSIAR-GWO has the second best optimization-finding ability compared to the other algorithms. Although the optimal values of all algorithms in F15 and F16 finally converge to close to the theoretical minimum, it can be found in Table 6 that MSIAR-GWO, ARO, DBO, and IGWO converge to the minimum value in each of the 20 individual tests in F15 and therefore achieve the smallest standard deviation and have a strong robustness. There is a large gap between the maximum and minimum values of WOA and AGWO in F16 indicates that these two algorithms are more likely to have difficulty escaping from falling into a local optimum during the optimization search process. In F18, the convergence accuracies of MSIAR-GWO and IGWO are higher than the other comparison algorithms, but by comparing the standard deviation evaluation indexes, it can be found that the MSIAR-GWO algorithm has an absolute advantage. As shown in Figure 8, Although most of the algorithms can find optimal solutions similar to the theoretical optimum on the fixed-dimension multi-peak benchmark test functions and there is not much difference in the ability to obtain the optimal results, MSIAR-GWO shows more stable search performance.

### 4.2. Wilcoxon Rank Sum Test

In order to further assess the performance of the MSIAR-GWO algorithm and to provide a more scientific analysis of the data obtained by the algorithms, a nonparametric statistical test was performed to determine the statistical significance of the results obtained by the MSIAR-GWO algorithm relative to the other algorithms; we used the Wilcoxon test at a 0.05 confidence level. If the *p*-value is less than 0.05, the two algorithms being tested are considered to be significantly different. On the contrary, the two algorithms being tested are considered to be not significantly different in terms of overall optimization results. When the *p*-value is NAN, this indicates that the two sets of samples are essentially the same. +/=/− indicates that the performance of the MSIAR-GWO algorithm is better, equal to, and worse than that of the compared algorithms. The Wilcoxon rank sum test results of the proposed MSIAR-GWO and the other seven algorithms are shown in Table 7, and Figure 9 shows that the proposed MSIAR-GWO algorithm outperforms the other algorithms in most benchmark functions, which further proves the effectiveness of MSIAR-GWO.

### 4.3. Mathematical Model and Simulation Results of Global Path Planning for Mobile Robots

#### 4.3.1. Environment Modeling

Environment modeling is the basis of global path planning for mobile robots, which directly affects the efficiency and effectiveness of a path planning algorithm. In this paper, the grid method is used to model the working environment of a mobile robot. The grid method divides the environment into regular grids, sets the side length of the grid to one unit length, expands and simplifies the irregular obstacles into obstacles represented by multiple regular grid cells, and can be divided into free grids and obstacle grids according to whether there are obstacles in the grid space. The value 1 is used to represent the obstacle area, and the obstacle grid is represented by a black grid. Furthermore, the value 0 is used to indicate the feasible region, the free grid is represented by a white grid, the starting point is denoted by S, and the ending point is denoted by T. For the modeling process, it first requires the generation of a random node in each column of the map between the starting and ending points, and then all nodes are placed in order on the path as gray wolf individuals. During this process, each gray wolf in the population is represented by a *d*-dimensional vector, for example, the ith gray wolf individual is represented by $X_{i}=\left(x_{1},x_{2},\ldots ,x_{d}\right)$, where *d* is the number of columns in the grid map. Secondly, after the generation of individuals, the path is made continuous and obstacle-free by interpolation. Moreover, the path is optimized by the direct connection test method to further reduce redundancy, eliminate unnecessary inflection points and bends, and obtain the shortest robot motion path, which makes it more efficient and concise and thus improves the work efficiency of the mobile robot, as shown in Figure 10. To further elaborate on the modeling process, the steps of path deduplication optimization are as follows:(1)Obtain an initial path consisting of a series of nodes.(2)Starting from the beginning of the path, connect to other nodes one by one by line segments.(3)Check whether the connecting line between the latter node and the former node is free of obstacles. If the area through which the connecting line passes is free of obstacles, remove all intermediate nodes between the two nodes.(4)After evaluating the first node and all subsequent nodes, repeat steps (2) and (3) starting from the next node until all pairs of nodes in the path have been checked.

Finally, the fitness function as shown in Equation (Equation 32) is constructed to calculate the fitness value of the individual path of the gray wolf and update the path shortest length and path shortest planning information, which are used as the objective function to be optimized by the algorithm to evaluate the optimization ability of the algorithm in path planning. The MSIAR-GWO algorithm is used to optimize the fitness function and find the shortest path.(32)$$Fit=O\left(Tx*Ty\right)+\sum_{i=1}^{N-1}\sqrt{{\left(Px_{i+1}-Px_{i}\right)}^{2}+{\left(Py_{i+1}-Py_{i}\right)}^{2}}$$
where *Tx* and *Ty* are the horizontal and vertical coordinates of the end point, *O* is the number of obstacles passed, *N* is the number of path nodes, and *Px* and *Py* are the horizontal and vertical coordinates of the path nodes, respectively.

The execution steps of the mobile robot path planning algorithm based on MSIAR-GWO are as follows:(1)Set the map environment parameters such as size, start position, and end position as well as the MSIAR-GWO algorithm parameters, population size, and maximum iteration number.(2)Initialize the population, calculate the fitness value corresponding to the individual gray wolf according to Equation (Equation 32), and select the leader gray wolf according to the fitness. Determine the path planning initial shortest path and the path shortest planning information.(3)Update the values of parameters a, A, and C via Equation (Equation 16), Equation (Equation 3), and Equation (Equation 4), respectively.(4)Update the individual position of each gray wolf according to Equation (Equation 20). If a < 1, the detour foraging strategy of Equation (Equation 24) and the greedy mechanism of Equation (Equation 28) are added to update the position of the current individual.(5)Through Equation (Equation 30), the inferior individuals are eliminated and their positioning is re-updated by using stochastic gravity dynamic opposition-based learning.(6)Calculate the path fitness function value to update the leader gray wolf, and update the path shortest length and path shortest planning information.(7)Determine whether the iteration termination condition is satisfied. If so, output the global shortest path length and path shortest planning information; otherwise, return to step 3 to continue optimization.(8)The algorithm ends and the best path planning result is output.

#### 4.3.2. Experimental Results

In order to verify the performance of the MSIAR-GWO algorithm in path planning more scientifically and objectively, the MSIAR-GWO algorithm and other seven algorithms are applied to the path planning problem of mobile robots under different complexity environments. Grids with dimensions 20 × 20, 30 × 30, and 40 × 40 are used, and the starting point and the end point of the robot’s path are located in the lower-left and upper-right corners, respectively. The eight different algorithms are repeated 20 times in the unused grid environment, and finally, the effectiveness of the algorithms is evaluated by calculating the optimal value, mean, and standard deviation of each algorithm.

The traditional graph search path planning algorithm based on a grid map has high efficiency in dealing with simple environments, but its efficiency and global optimization ability is limited in complex environments, such as dense obstacles and multiple path choices. Compared with the traditional A* algorithm through simulation experiments, the results in Table 8 and Figure 11 show that the optimal path length of the MSIAR-GWO algorithm is reduced by 4.22% and 2.54%, respectively, compared with that of the A* algorithm under the 20 × 20 and 40 × 40 grid maps, indicating that the MSIAR-GWO algorithm is competitive in solving the shortest path problem. In particular, it shows stronger global optimization ability in a complex environment.

In the results of path planning under 20 × 20 and 30 × 30 raster maps in Table 9 and Table 10, although the MSIAR-GWO and ARO algorithms converge to the same optimal value, the MSIAR-GWO algorithm outperforms the ARO algorithm in terms of both mean and standard deviation. Figure 12 and Figure 13 respectively show the average path and optimal path obtained by each algorithm in the 20 × 20 and 30 × 30 grid environment, and it can be found that the improved algorithms show strong stability and robustness, which is attributed to the fact that the parameter optimization by reinforcement learning can be very effective in terms of the nonlinear convergence factor. The nonlinear convergence factor can well control the process of global exploration and local exploitation of wolves. In the 20 × 20 raster path, the path length of WOA planning is high and the standard deviation is large, and the path planning is unstable. The average path length of the path planning MSIAR-GWO algorithm was reduced by 4.33%, 0.73%, 4.18%, 11.54%, 0.58%, 6.54%, and 3.57% compared to the GWO, ARO, DBO, WOA, IGWO, AGWO, and RSMGWO algorithms, respectively; in 30 × 30 raster paths, it was reduced by 17.77%, 4.27%, 11.99%, 14.52%, 6.09%, 15.02%, and 14.59%; It can be seen from Table 11 and Figure 14 that in the 40 × 40 high-dimensional complex environment, the optimal value, mean value and standard deviation show absolute advantages over other algorithms. Compared with GWO, ARO, DBO, WOA, IGWO, AGWO and RSMGWO algorithms, the average path length of the proposed algorithm is reduced by 22.88%, 4.47%, 7.10%, 8.32%, 19.40%, 22.65% and 21.97%, respectively. From the shortest paths obtained by each algorithm in Figure 15, Figure 16 and Figure 17 in different environments, it can be seen that the MSIAR-GWO algorithm can obtain shorter and smoother paths. In the average convergence curve of path planning from Figure 18, Figure 19 and Figure 20, although the MSIAR-GWO algorithm is not the fastest converging algorithm at the beginning of the iteration, it achieves higher convergence accuracy compared to other algorithms at the later stages of the iterations. With the higher dimension of the raster map and the more complex environment, the other algorithms have insufficient searching ability in the late iterations, which leads the algorithm to fall into the local optimum. By combining and improving the detour foraging behavior of the rabbit pack as well as eliminating and relocating the inferior individuals of the wolf pack, the diversity of the population is enriched, and the algorithm’s ability of global search and escaping from the local optimum is improved, which enhances the ability of the mobile robot to adequately search for the paths. The comparison results show that the proposed improved algorithm is more effective in solving the robot path planning problem.

## 5. Conclusions and Prospects

Aiming at the limitations of the basic gray wolf optimization algorithm, such as slow convergence speed, insufficient solution accuracy, and proneness to premature maturity, which lead to inefficiency when applied to path planning, a multi-strategy improved gray wolf optimization algorithm based on reinforcement learning (MSIAR-GWO) is proposed. First, intelligent selection of adjustable parameters is performed through reinforcement learning to change the limitations of traditional parameter tuning methods that usually rely on manual experience and repeated trials. Second, an adaptive position-updating strategy based on detour foraging and dynamic weights is proposed to dynamically adjust the proportional weights according to the adaptability of the leadership hierarchy to enhance the role of dominant individuals in the leadership to guide and accelerate the convergence speed of the algorithm. By combining the detour foraging strategy of ARO to fully maximize the advantages of the two algorithms, the method maintains the powerful development capability of GWO and can utilize the efficient exploration capability of ARO, balances the search performance of GWO, and thus improves the convergence accuracy of the algorithm. Furthermore, the detour foraging strategy is improved by adding Levy flight and Brownian motion, so that the search can not only cover the whole solution space to enhance the global search capability but also conduct fine search near the high-quality solution to enhance the algorithm’s solution accuracy, thus realizing the balance between exploitation and exploration. Finally, stochastic center-of-gravity dynamic reverse learning is performed on the inferior individuals in the wolf pack, which increases the diversity of wolf pack individuals, thus keeping the algorithm from falling into the local optimum. On this basis, both simulation experiments in 18 test functions and the Wilcoxon rank sum test well verify the optimization performance of MSIAR-GWO, which obtains better optimal solutions in the vast majority of test functions and significantly improves the method in terms of solution accuracy compared with other GWO variants and different types of algorithms. The metrics such as mean and standard deviation show that the proposed algorithm performs more consistently in terms of average optimization performance and has strong robustness. Then, simulation experiments on path planning were conducted using MSIAR-GWO in three different complexity scenarios, and the experimental results show that MSIAR-GWO is able to plan shorter and safer paths in various scenarios and at the same time exhibits stronger robustness and effectiveness compared to other algorithms involved in the experiments. Although the proposed algorithm achieves better results, in this study, we only consider global path planning for mobile robots in static environments with path length and safety as the optimization objectives. In future work, we will further explore the application of the algorithm in dynamic obstacle environments and introduce more path evaluation metrics according to practical needs.

## Figures and Tables

**Figure 1 sensors-25-00892-f001:**
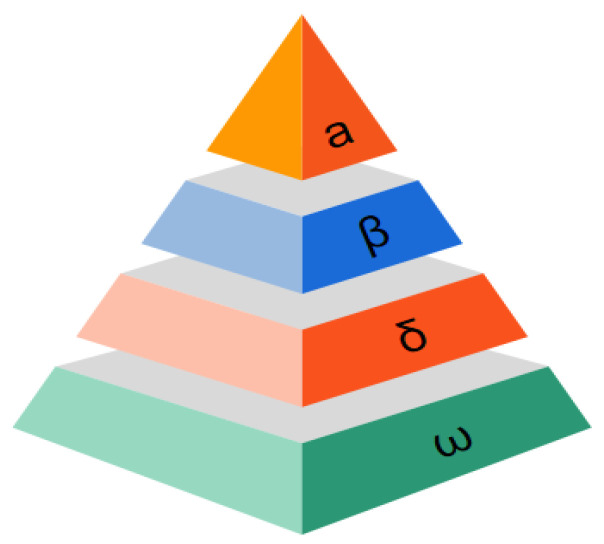
Social hierarchy pyramid of the gray wolf population.

**Figure 2 sensors-25-00892-f002:**
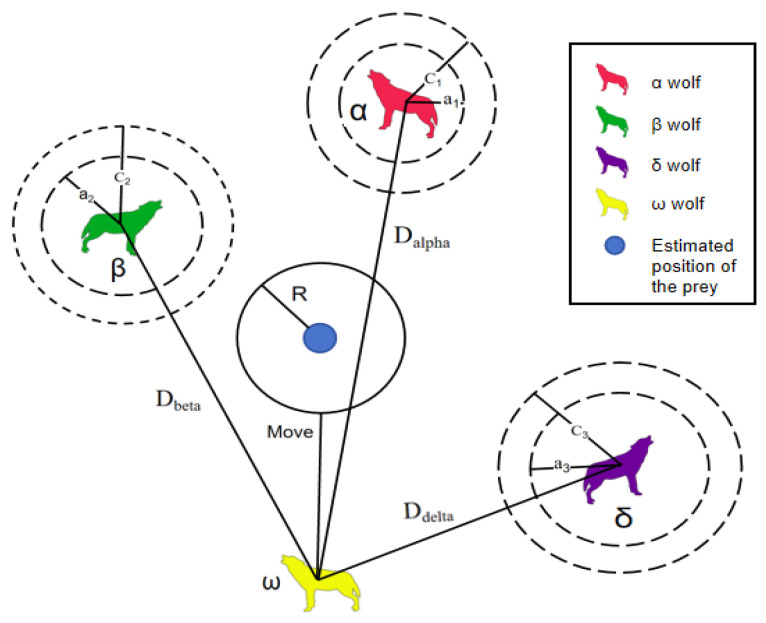
Schematic diagram of the mechanism for updating the location of the gray wolf population.

**Figure 3 sensors-25-00892-f003:**
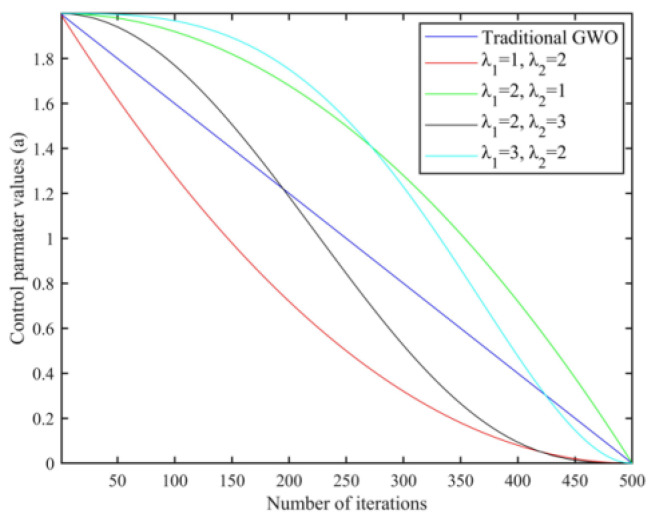
Comparison of the convergence factors for different $\lambda_{1}$ and $\lambda_{2}$.

**Figure 4 sensors-25-00892-f004:**
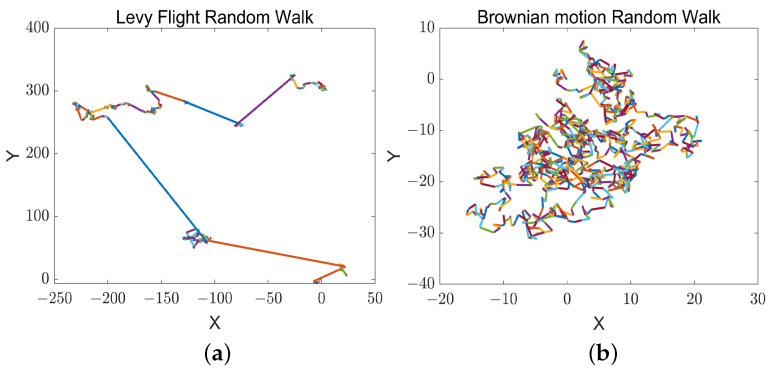
Trajectory distribution diagram of Levy flight and Brownian motion. (**a**) Levy flight trajectory distribution diagram; (**b**) Brownian motion trajectory distribution diagram.

**Figure 5 sensors-25-00892-f005:**
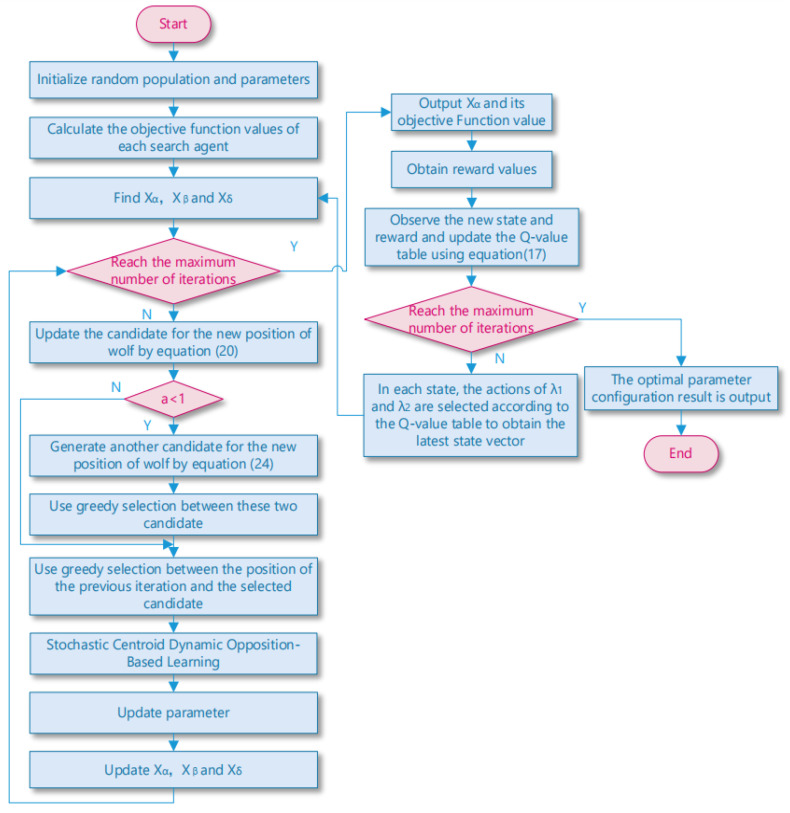
Flowchart of MSIAR-GWO.

**Figure 6 sensors-25-00892-f006:**
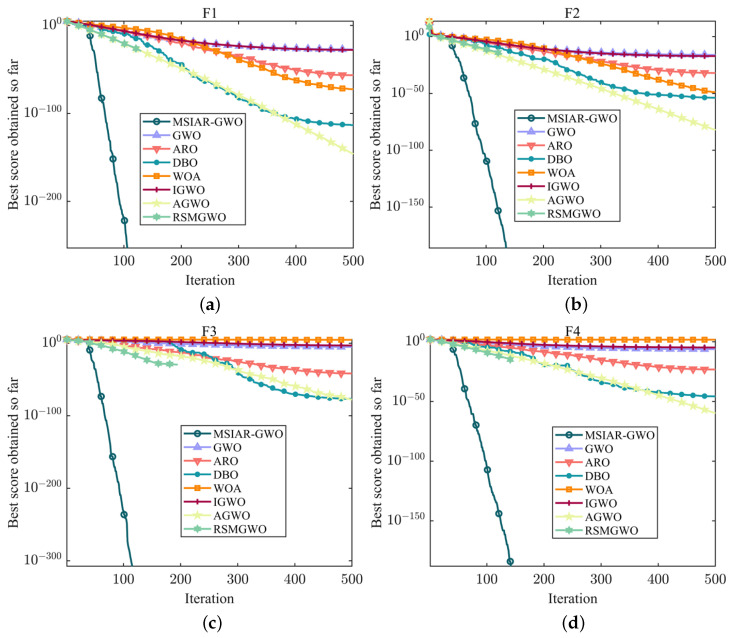
Convergence curves of different algorithms on a unimodal benchmark test function. (**a**) The convergence curve of F1; (**b**) The convergence curve of F2; (**c**) The convergence curve of F3; (**d**) The convergence curve of F4; (**e**) The convergence curve of F5; (**f**) The convergence curve of F6; (**g**) The convergence curve of F7.

**Figure 7 sensors-25-00892-f007:**
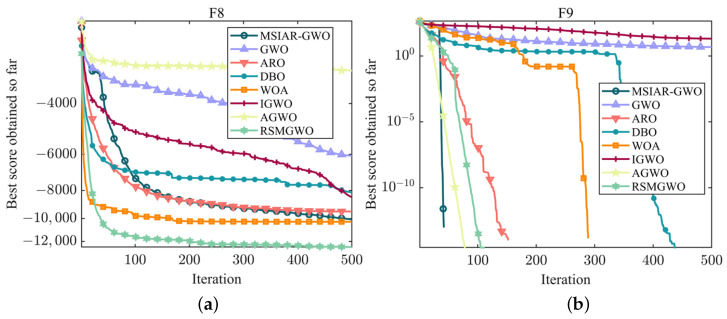
Convergence curves of different algorithms on multi-modal benchmark test functions. (**a**) The convergence curve of F8; (**b**) The convergence curve of F9; (**c**) The convergence curve of F10; (**d**) The convergence curve of F11; (**e**) The convergence curve of F12; (**f**) The convergence curve of F13.

**Figure 8 sensors-25-00892-f008:**
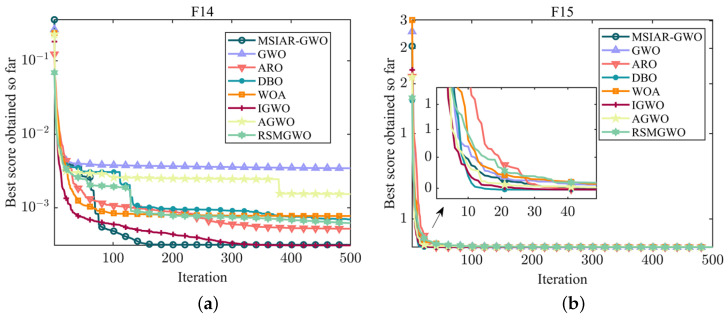
Convergence curves of different algorithms on fixed-dimensional multi-modal benchmark test functions. (**a**) The convergence curve of F14; (**b**) The convergence curve of F15; (**c**) The convergence curve of F16; (**d**) The convergence curve of F17; (**e**) The convergence curve of F18.

**Figure 9 sensors-25-00892-f009:**
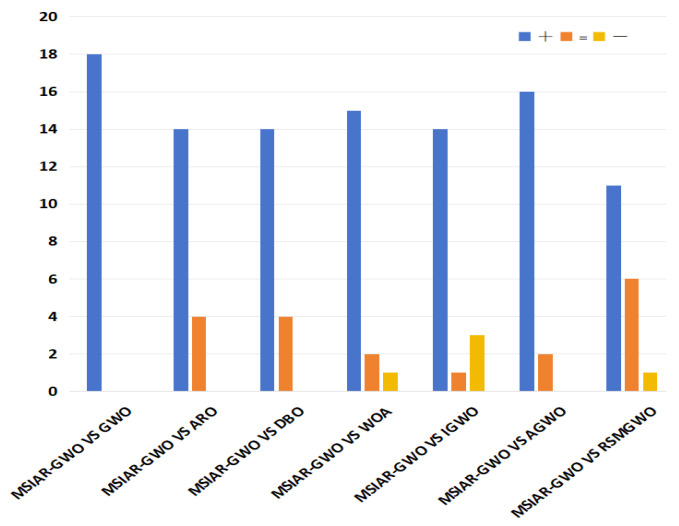
Performance comparison between MSIAR-GWO and other algorithms.

**Figure 10 sensors-25-00892-f010:**
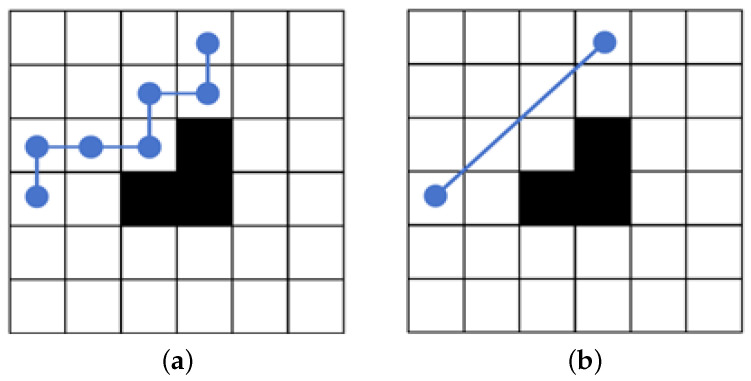
Comparison of paths before and after de-redundancy optimization. (**a**) The original path without de-redundancy optimization; (**b**) The path after the de-redundancy optimization.

**Figure 11 sensors-25-00892-f011:**
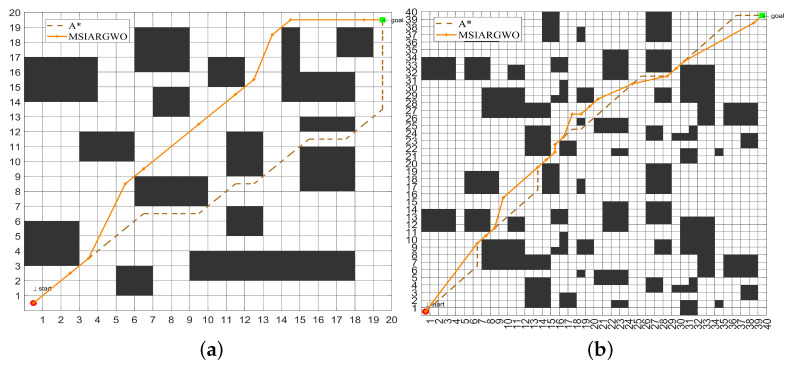
Comparison experiment with the traditional A* algorithm in raster environment. (**a**) Optimal path of MSIAR-GWO algorithm and A* algorithm under 20 × 20 raster map; (**b**) Optimal path of MSIAR-GWO algorithm and A* algorithm under 40 × 40 raster map.

**Figure 12 sensors-25-00892-f012:**
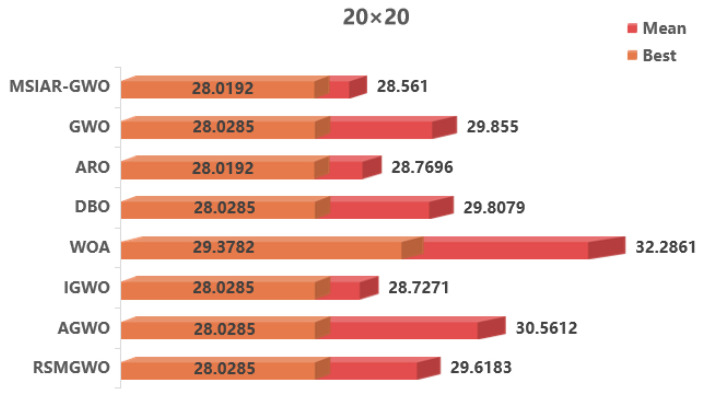
Average path and best path in a 20 × 20 raster environment.

**Figure 13 sensors-25-00892-f013:**
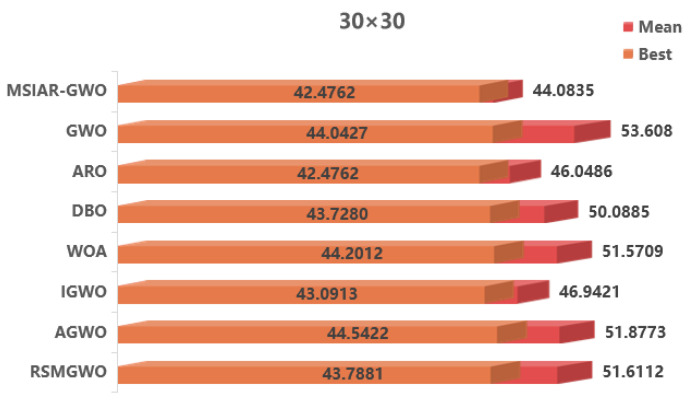
Average path and best path in a 30 × 30 raster environment.

**Figure 14 sensors-25-00892-f014:**
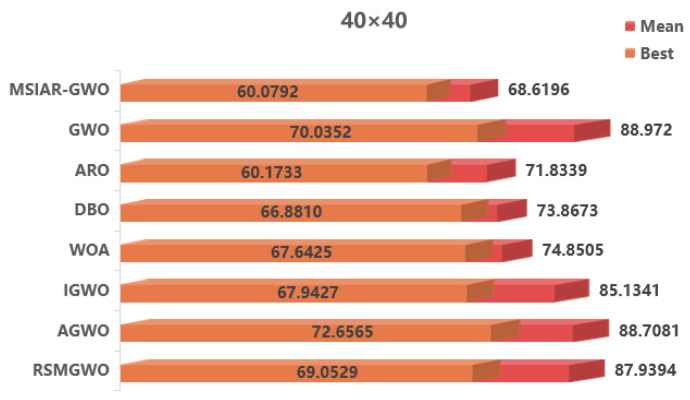
Average path and best path in a 40 × 40 raster environment.

**Figure 15 sensors-25-00892-f015:**
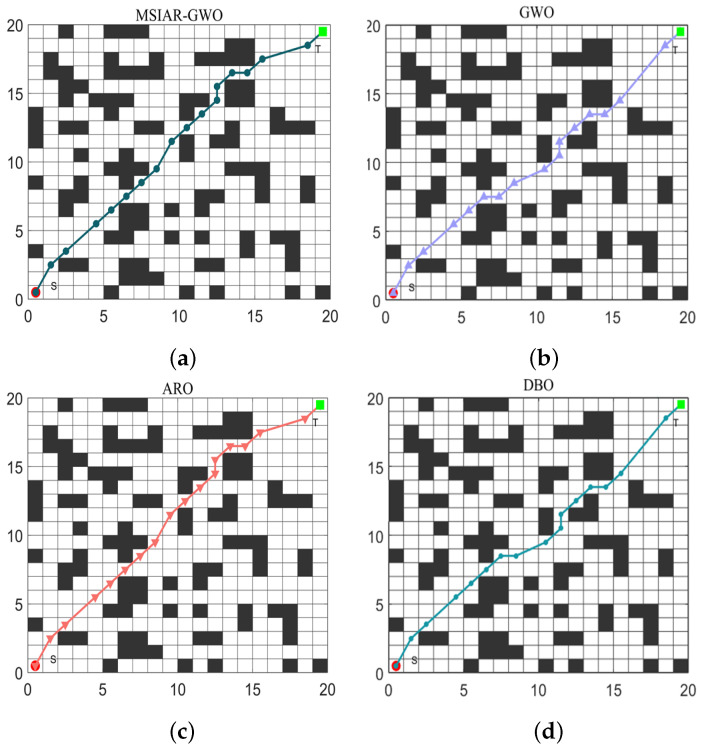
Shortest path planning graph for 8 algorithms in 20 × 20 raster environment. (**a**) MSIAR-GWO; (**b**) GWO; (**c**) ARO; (**d**) DBO; (**e**) WOA; (**f**) IGWO; (**g**) AGWO; (**h**) RSMGWO.

**Figure 16 sensors-25-00892-f016:**
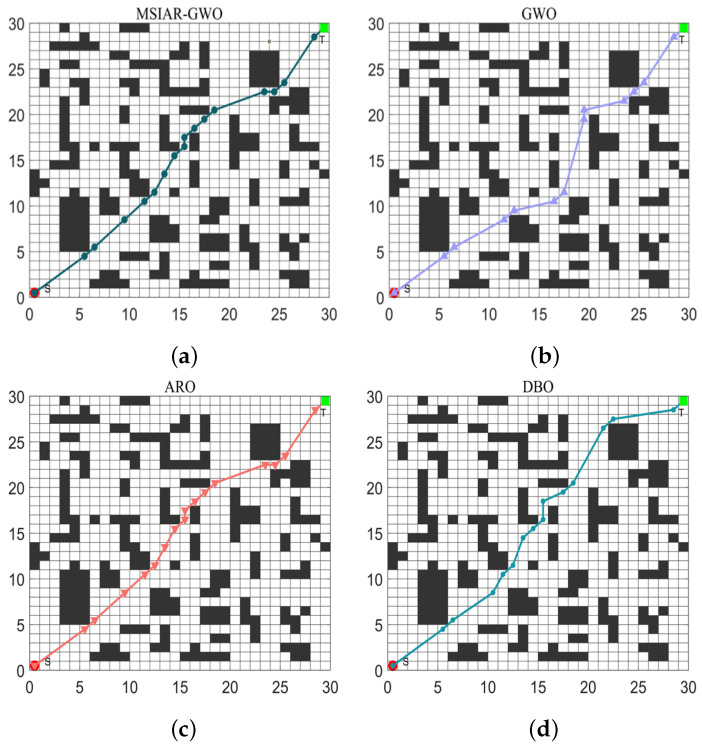
Graph of shortest path planning for 8 algorithms in 30 × 30 raster environment. (**a**) MSIAR-GWO; (**b**) GWO; (**c**) ARO; (**d**) DBO; (**e**) WOA; (**f**) IGWO; (**g**) AGWO; (**h**) RSMGWO.

**Figure 17 sensors-25-00892-f017:**
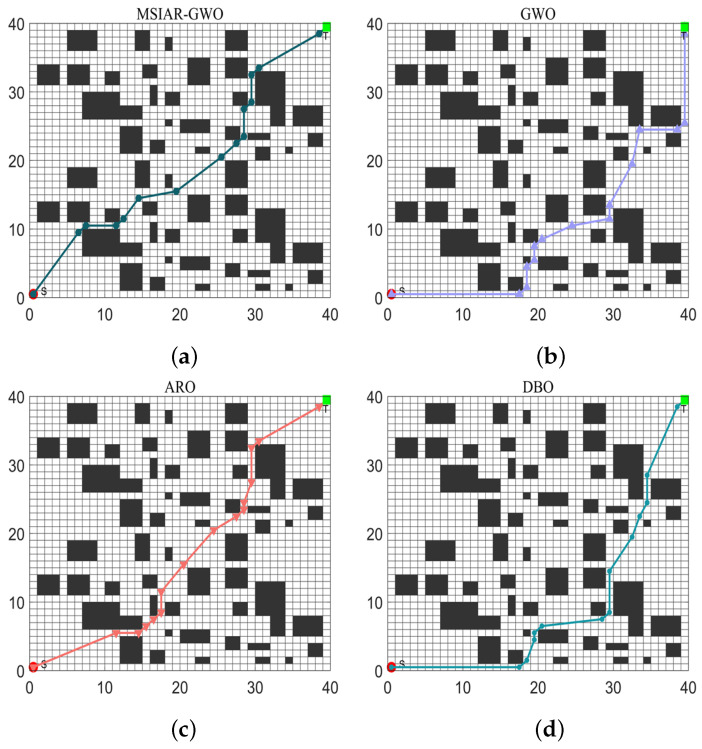
Shortest path planning for 8 algorithms in 40 × 40 raster environment. (**a**) MSIAR-GWO; (**b**) GWO; (**c**) ARO; (**d**) DBO; (**e**) WOA; (**f**) IGWO; (**g**) AGWO; (**h**) RSMGWO.

**Figure 18 sensors-25-00892-f018:**
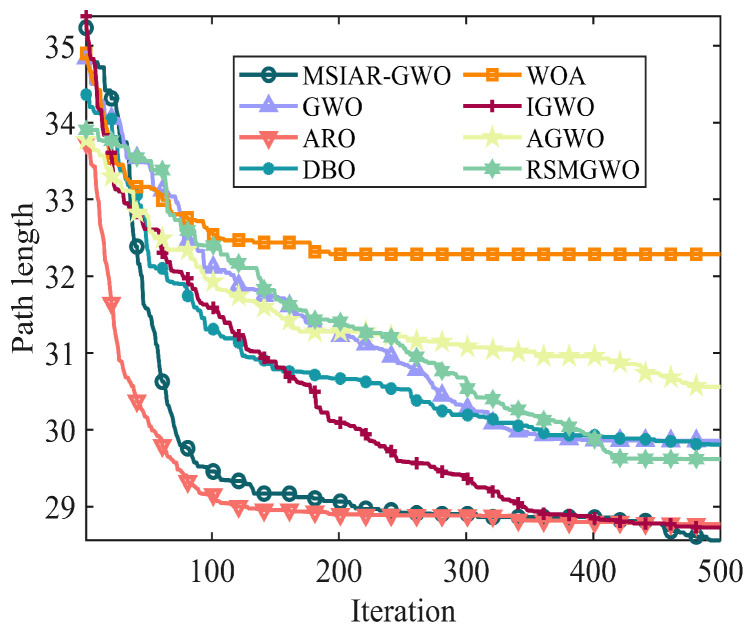
Average convergence curves in a 20 × 20 grid environment.

**Figure 19 sensors-25-00892-f019:**
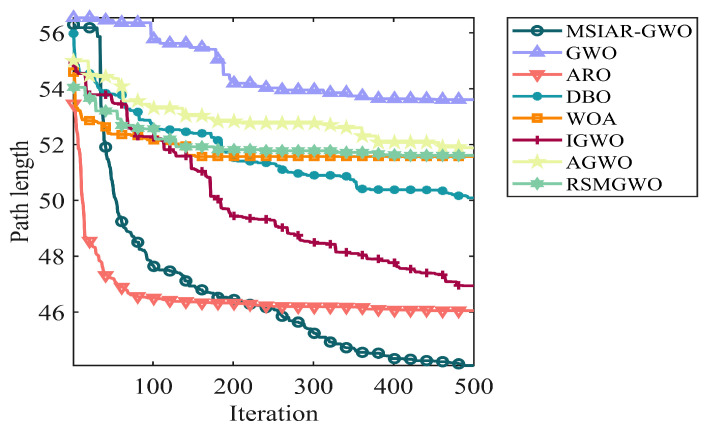
Average convergence curves in a 30 × 30 grid environment.

**Figure 20 sensors-25-00892-f020:**
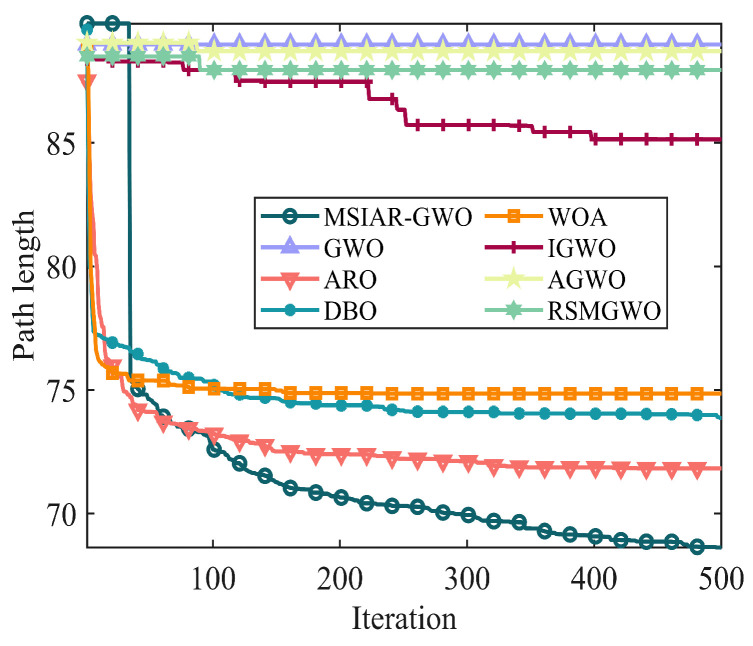
Average convergence curves in a 40 × 40 grid environment.

**Table 1 sensors-25-00892-t001:** Single-peak benchmark test functions.

Function	Dim	Range	$f_{\min}$
$F1\left(x\right)=\sum_{i=1}^{d}x_{i}^{2}$	30	[−100, 100]	0
$F2\left(x\right)=\sum_{i=1}^{d}\left|x_{i}\right|+\prod_{i=1}^{d}\left|x_{i}\right|$	30	[−10, 10]	0
$F3\left(x\right)=\sum_{i=1}^{d}{\left(\sum_{j=1}^{i}x_{j}\right)}^{2}$	30	[−100, 100]	0
$F4\left(x\right)=\max_{i}\left\{\left|x_{i}\right|,1\leq i\leq d\right\}$	30	[−100, 100]	0
$F5\left(x\right)=\sum_{i=1}^{d-1}\left[100{\left(x_{i+1}-x_{i}^{2}\right)}^{2}+{\left(x_{i}-1\right)}^{2}\right]$	30	[−30, 30]	0
$F6\left(x\right)=\sum_{i=1}^{d}{\left(\left[x_{i}+5\right]\right)}^{2}$	30	[−100, 100]	0
$F7\left(x\right)=\sum_{i=1}^{d}i\cdot x_{i}^{4}+\mathrm{rand}\left[0,1\right)$	30	[−1.28, 1.28]	0

**Table 2 sensors-25-00892-t002:** Multi-peak benchmark test functions.

Function	Dim	Range	$f_{\min}$
$F8\left(x\right)=\sum_{i=1}^{d}-x_{i}\sin\left(\sqrt{\left|x_{i}\right|}\right)$	30	[−500, 500]	−418.9829×d
$F9\left(x\right)=\sum_{i=1}^{d}\left[x_{i}^{2}-10\cos\left(2x_{i}\pi \right)+10\right]$	30	[−5.12, 5.12]	0
$\begin{array}{cc} & F10\left(x\right)=-20\exp\left(-0.2\sqrt{\frac{1}{d}\sum_{i=1}^{d}x_{i}^{2}}\right)+20+e\\ & -\exp\left[\frac{1}{d}\sum_{i=1}^{d}\cos\left(2x_{i}\pi \right)\right] \end{array}$	30	[−32, 32]	0
$F11\left(x\right)=\frac{1}{4000}\sum_{i=1}^{d}x_{i}^{2}-\prod_{i=1}^{d}\cos\left(\frac{x_{i}}{\sqrt{i}}\right)+1$	30	[−600, 600]	0
$\begin{array}{cc} & F12\left(x\right)=\sum_{i=1}^{d}u\left(x_{i},10,100,4\right)+\frac{\pi }{d}\left\{10\sin\left(\pi y_{i}\right)\right.\\ & +{\left(y_{i}-1\right)}^{2}+\sum_{i=1}^{d-1}{\left(y_{i}-1\right)}^{2}\left[1+10\sin^{2}\left(\pi y_{i+1}+1\right)\right]\\ & y_{i}=1+\frac{x_{i}+1}{4}\\ & u\left(x_{i},a,k,m\right)=\left\{\begin{array}{cc} k{\left(x_{i}-a\right)}^{m} & x_{i}>a\\ 0 & -a<x_{i}<a\\ k{\left(-x_{i}-a\right)}^{m} & x_{i}<-a \end{array}\right. \end{array}$	30	[−50, 50]	0
$\begin{array}{cc} & F13\left(x\right)=0.1\times \left\{\sin^{2}\left(3\pi x_{1}\right)+{\left(x_{d}-1\right)}^{2}\left[1+\sin^{2}\left(2\pi x_{n}\right)\right]\right.\\ & \left.+\sum_{i=1}^{d}{\left(x_{i}-1\right)}^{2}\left[1+\sin^{2}\left(1+3\pi x_{i}\right)\right]\right\}+\sum_{i=1}^{d}u\left(x_{i},5,100,4\right) \end{array}$	30	[−50, 50]	0

**Table 3 sensors-25-00892-t003:** Fixed dimensional multi-peak benchmarking functions.

Function	Dim	Range	$f_{\min}$
$F14\left(x\right)=\sum_{i=1}^{11}{\left[a_{i}-\frac{x_{1}\left(b_{i}^{2}+b_{i}x_{2}\right)}{b_{i}^{2}+b_{i}x_{3}+x_{4}}\right]}^{2}$	4	[−5, 5]	0.0003
$\begin{array}{cc} & F15\left(x\right)=10+10\times \left(1-\frac{0.125}{\pi }\right)\cos\left(x_{1}\right)\\ & +{\left(x_{2}-\frac{5.1}{4\pi^{2}}x_{1}^{2}+\frac{5}{\pi }x_{1}-6\right)}^{2} \end{array}$	2	[−5, 5]	0.398
$\begin{array}{cc} & F16\left(x\right)=\left[1+{\left(1+x_{1}+x_{2}\right)}^{2}\left(19-14x_{1}+3x_{1}^{2}-14x_{2}\right.\right.\\ & \left.\left.+6x_{1}x_{2}+3x_{2}^{2}\right)\right]\times \left[30+{\left(2x_{1}-3x_{2}\right)}^{2}\times \left(18-32x_{1}\right.\right.\\ & \left.\left.-36x_{1}x_{2}+48x_{2}+27x_{2}^{2}\right)\right] \end{array}$	2	[−2, 2]	3
$F17\left(x\right)=-\sum_{i=1}^{4}c_{i}\exp\left(-\sum_{j=1}^{6}a_{ij}{\left(x_{j}-p_{ij}\right)}^{2}\right)$	6	[0, 1]	−3.32
$F18\left(x\right)=-\sum_{i=1}^{10}{\left[\left(x-a_{i}\right){\left(x-a_{i}\right)}^{T}+c_{i}\right]}^{-1}$	4	[0, 10]	−10.5363

**Table 4 sensors-25-00892-t004:** Experimental comparison results of MSIAR-GWO and other advanced algorithms on single-peak benchmark test functions.

Function	Algorithm	Best	Mean	Worst	St. dev
F1	MSIAR-GWO	0	0	0	0
	GWO	$1.3821\times 10^{-29}$	$6.2807\times 10^{-28}$	$6.8549\times 10^{-27}$	$1.4913\times 10^{-27}$
	ARO	$4.4226\times 10^{-71}$	$3.0883\times 10^{-57}$	$5.5272\times 10^{-56}$	$1.2321\times 10^{-56}$
	DBO	$8.6336\times 10^{-154}$	$4.7596\times 10^{-114}$	$9.5140\times 10^{-113}$	$2.1273\times 10^{-113}$
	WOA	$1.1821\times 10^{-91}$	$2.8898\times 10^{-73}$	$5.6276\times 10^{-72}$	$1.2570\times 10^{-72}$
	IGWO	$6.4995\times 10^{-30}$	$1.6309\times 10^{-28}$	$8.6295\times 10^{-28}$	$2.1244\times 10^{-28}$
	AGWO	$2.0842\times 10^{-153}$	$6.3503\times 10^{-147}$	$4.8579\times 10^{-146}$	$1.2659\times 10^{-146}$
	RSMGWO	0	0	0	0
F2	MSIAR-GWO	0	0	0	0
	GWO	$9.4259\times 10^{-18}$	$9.3570\times 10^{-17}$	$2.3449\times 10^{-16}$	$5.5576\times 10^{-17}$
	ARO	$1.0296\times 10^{-37}$	$1.0402\times 10^{-32}$	$1.3724\times 10^{-31}$	$3.0770\times 10^{-32}$
	DBO	$9.9074\times 10^{-84}$	$1.7019\times 10^{-54}$	$3.3823\times 10^{-53}$	$7.5606\times 10^{-54}$
	WOA	$4.0003\times 10^{-58}$	$1.6589\times 10^{-49}$	$3.2951\times 10^{-48}$	$7.3655\times 10^{-49}$
	IGWO	$1.2891\times 10^{-18}$	$7.7487\times 10^{-18}$	$2.9549\times 10^{-17}$	$6.8358\times 10^{-18}$
	AGWO	$2.0074\times 10^{-85}$	$5.4538\times 10^{-83}$	$5.0529\times 10^{-82}$	$1.2493\times 10^{-82}$
	RSMGWO	0	0	0	0
F3	MSIAR-GWO	0	0	0	0
	GWO	$3.6128\times 10^{-08}$	$7.8476\times 10^{-06}$	$7.1567\times 10^{-05}$	$1.6877\times 10^{-05}$
	ARO	$4.9157\times 10^{-57}$	$2.0089\times 10^{-42}$	$2.3564\times 10^{-41}$	$5.4775\times 10^{-42}$
	DBO	$6.8009\times 10^{-152}$	$2.1176\times 10^{-77}$	$3.3804\times 10^{-76}$	$7.6985\times 10^{-77}$
	WOA	$1.7750\times 10^{+04}$	$4.4689\times 10^{+04}$	$7.6132\times 10^{+04}$	$1.3097\times 10^{+04}$
	IGWO	$9.0625\times 10^{-06}$	$7.5310\times 10^{-04}$	$6.2795\times 10^{-03}$	$1.5675\times 10^{-03}$
	AGWO	$2.4176\times 10^{-90}$	$2.5827\times 10^{-78}$	$4.1985\times 10^{-77}$	$9.5077\times 10^{-78}$
	RSMGWO	0	0	0	0
F4	MSIAR-GWO	0	0	0	0
	GWO	$3.8929\times 10^{-08}$	$5.9376\times 10^{-07}$	$2.1950\times 10^{-06}$	$5.3346\times 10^{-07}$
	ARO	$5.4613\times 10^{-29}$	$5.9914\times 10^{-24}$	$6.7252\times 10^{-23}$	$1.6191\times 10^{-23}$
	DBO	$1.0807\times 10^{-77}$	$1.7637\times 10^{-46}$	$3.4543\times 10^{-45}$	$7.7171\times 10^{-46}$
	WOA	$6.7361\times 10^{-01}$	$5.5203\times 10^{+01}$	$8.9990\times 10^{+01}$	$2.8956\times 10^{+01}$
	IGWO	$2.7453\times 10^{-06}$	$1.5898\times 10^{-05}$	$3.9046\times 10^{-05}$	$1.1701\times 10^{-05}$
	AGWO	$1.2163\times 10^{-64}$	$1.6983\times 10^{-60}$	$1.8937\times 10^{-59}$	$4.5022\times 10^{-60}$
	RSMGWO	0	0	0	0
F5	MSIAR-GWO	$1.6460\times 10^{-09}$	$2.0653\times 10^{-04}$	$6.8179\times 10^{-04}$	$2.3260\times 10^{-04}$
	GWO	$2.6143\times 10^{+01}$	$2.7017\times 10^{+01}$	$2.8003\times 10^{+01}$	$5.5363\times 10^{-01}$
	ARO	$1.6546\times 10^{-02}$	$2.0078\times 10^{-01}$	$1.1664\times 10^{00}$	$2.7292\times 10^{-01}$
	DBO	$2.5305\times 10^{+01}$	$2.5743\times 10^{+01}$	$2.6168\times 10^{+01}$	$2.0202\times 10^{-01}$
	WOA	$2.6952\times 10^{+01}$	$2.7888\times 10^{+01}$	$2.8729\times 10^{+01}$	$4.3934\times 10^{-01}$
	IGWO	$2.3515\times 10^{+01}$	$2.4242\times 10^{+01}$	$2.4858\times 10^{+01}$	$3.7595\times 10^{-01}$
	AGWO	$2.7180\times 10^{+01}$	$2.7815\times 10^{+01}$	$2.8831\times 10^{+01}$	$5.6113\times 10^{-01}$
	RSMGWO	$2.6253\times 10^{+01}$	$2.7181\times 10^{+01}$	$2.7930\times 10^{+01}$	$4.0670\times 10^{-01}$
F6	MSIAR-GWO	$5.5632\times 10^{-24}$	$3.4325\times 10^{-22}$	$1.9650\times 10^{-21}$	$5.0581\times 10^{-22}$
	GWO	$2.5036\times 10^{-01}$	$7.7503\times 10^{-01}$	$1.4354\times 10^{00}$	$3.1206\times 10^{-01}$
	ARO	$3.8359\times 10^{-04}$	$1.1158\times 10^{-03}$	$2.1466\times 10^{-03}$	$4.7279\times 10^{-04}$
	DBO	$1.9957\times 10^{-05}$	$8.9116\times 10^{-03}$	$1.4903\times 10^{-01}$	$3.3450\times 10^{-02}$
	WOA	$5.8497\times 10^{-02}$	$4.2943\times 10^{-01}$	$8.4182\times 10^{-01}$	$2.3436\times 10^{-01}$
	IGWO	$4.0195\times 10^{-05}$	$2.3054\times 10^{-02}$	$2.4955\times 10^{-01}$	$7.1018\times 10^{-02}$
	AGWO	$2.5563\times 10^{00}$	$3.3558\times 10^{00}$	$3.7923\times 10^{00}$	$3.9519\times 10^{-01}$
	RSMGWO	$2.2949\times 10^{-06}$	$7.5759\times 10^{-05}$	$3.0669\times 10^{-04}$	$8.5746\times 10^{-05}$
F7	MSIAR-GWO	$5.9229\times 10^{-06}$	$7.0305\times 10^{-05}$	$2.9673\times 10^{-04}$	$6.8959\times 10^{-05}$
	GWO	$3.1165\times 10^{-04}$	$2.2913\times 10^{-03}$	$5.7064\times 10^{-03}$	$1.2168\times 10^{-03}$
	ARO	$7.0889\times 10^{-05}$	$6.2082\times 10^{-04}$	$2.9726\times 10^{-03}$	$6.2281\times 10^{-04}$
	DBO	$8.4590\times 10^{-05}$	$1.2396\times 10^{-03}$	$3.3164\times 10^{-03}$	$9.2752\times 10^{-04}$
	WOA	$8.5273\times 10^{-05}$	$4.5360\times 10^{-03}$	$1.3377\times 10^{-02}$	$4.0887\times 10^{-03}$
	IGWO	$5.8303\times 10^{-04}$	$2.3365\times 10^{-03}$	$3.9505\times 10^{-03}$	$9.8040\times 10^{-04}$
	AGWO	$2.0011\times 10^{-05}$	$2.3057\times 10^{-04}$	$6.9811\times 10^{-04}$	$2.1565\times 10^{-04}$
	RSMGWO	$5.9145\times 10^{-05}$	$3.2236\times 10^{-04}$	$1.1695\times 10^{-03}$	$2.7439\times 10^{-04}$

**Table 5 sensors-25-00892-t005:** Experimental comparison results of MSIAR-GWO and other advanced algorithms on multi-peak benchmark test functions.

Function	Algorithm	Best	Mean	Worst	St. dev
F8	MSIAR-GWO	−$1.0836\times 10^{+04}$	−$1.0090\times 10^{+04}$	−$9.2636\times 10^{+03}$	$5.4178\times 10^{+02}$
	GWO	−$7.9404\times 10^{+03}$	−$6.0610\times 10^{+03}$	−$5.0491\times 10^{+03}$	$7.1838\times 10^{+02}$
	ARO	−$1.0554\times 10^{+04}$	−$9.4584\times 10^{+03}$	−$8.8839\times 10^{+03}$	$4.2706\times 10^{+02}$
	DBO	−$1.1916\times 10^{+04}$	−$8.1084\times 10^{+03}$	−$6.2456\times 10^{+03}$	$1.9015\times 10^{+03}$
	WOA	−$1.2569\times 10^{+04}$	−$1.0262\times 10^{+04}$	−$8.5139\times 10^{+03}$	$1.7111\times 10^{+03}$
	IGWO	−$1.0374\times 10^{+04}$	−$8.4198\times 10^{+03}$	−$5.5540\times 10^{+03}$	$1.3328\times 10^{+03}$
	AGWO	−$3.6820\times 10^{+03}$	−$3.0770\times 10^{+03}$	−$2.4996\times 10^{+03}$	$3.2746\times 10^{+02}$
	RSMGWO	−$1.2569\times 10^{+04}$	−$1.2564\times 10^{+04}$	−$1.2543\times 10^{+04}$	$7.6103\times 10^{00}$
F9	MSIAR-GWO	0	0	0	0
	GWO	$5.6843\times 10^{-14}$	$4.7832\times 10^{00}$	$1.6622\times 10^{+01}$	$5.2198\times 10^{00}$
	ARO	0	0	0	0
	DBO	0	0	0	0
	WOA	0	0	0	0
	IGWO	$8.2109\times 10^{00}$	$2.0779\times 10^{+01}$	$4.9583\times 10^{+01}$	$1.0060\times 10^{+01}$
	AGWO	0	0	0	0
	RSMGWO	0	0	0	0
F10	MSIAR-GWO	$8.8818\times 10^{-16}$	$8.8818\times 10^{-16}$	$8.8818\times 10^{-16}$	0
	GWO	$6.8390\times 10^{-14}$	$9.4680\times 10^{-14}$	$1.2168\times 10^{-13}$	$1.4369\times 10^{-14}$
	ARO	$8.8818\times 10^{-16}$	$8.8818\times 10^{-16}$	$8.8818\times 10^{-16}$	0
	DBO	$8.8818\times 10^{-16}$	$8.8818\times 10^{-16}$	$8.8818\times 10^{-16}$	0
	WOA	$8.8818\times 10^{-16}$	$4.2633\times 10^{-15}$	$7.9936\times 10^{-15}$	$2.9330\times 10^{-15}$
	IGWO	$5.0626\times 10^{-14}$	$6.4837\times 10^{-14}$	$8.6153\times 10^{-14}$	$9.2930\times 10^{-15}$
	AGWO	$4.4409\times 10^{-15}$	$4.6185\times 10^{-15}$	$7.9936\times 10^{-15}$	$7.9441\times 10^{-15}$
	RSMGWO	$8.8818\times 10^{-16}$	$4.2633\times 10^{-15}$	$7.9936\times 10^{-15}$	$1.3999\times 10^{-15}$
F11	MSIAR-GWO	0	0	0	0
	GWO	0	$6.6876\times 10^{-03}$	$4.0185\times 10^{-02}$	$1.0935\times 10^{-02}$
	ARO	0	0	0	0
	DBO	0	0	0	0
	WOA	0	0	0	0
	IGWO	0	$3.2040\times 10^{-03}$	$2.7061\times 10^{-02}$	$7.3701\times 10^{-03}$
	AGWO	0	0	0	0
	RSMGWO	0	0	0	0
F12	MSIAR-GWO	$4.0323\times 10^{-18}$	$7.4258\times 10^{-17}$	$2.2160\times 10^{-16}$	$6.6010\times 10^{-17}$
	GWO	$2.6330\times 10^{-02}$	$3.9945\times 10^{-02}$	$5.9463\times 10^{-02}$	$9.7664\times 10^{-03}$
	ARO	$6.3051\times 10^{-06}$	$1.1974\times 10^{-04}$	$1.1507\times 10^{-03}$	$2.4711\times 10^{-04}$
	DBO	$2.6614\times 10^{-07}$	$6.6102\times 10^{-04}$	$6.7746\times 10^{-03}$	$1.8243\times 10^{-03}$
	WOA	$6.2320\times 10^{-03}$	$3.1903\times 10^{-02}$	$2.2334\times 10^{-01}$	$4.6290\times 10^{-02}$
	IGWO	$3.9895\times 10^{-06}$	$6.5986\times 10^{-04}$	$6.7869\times 10^{-03}$	$2.0114\times 10^{-03}$
	AGWO	$1.9715\times 10^{-01}$	$3.1373\times 10^{-01}$	$7.5955\times 10^{-01}$	$1.2646\times 10^{-01}$
	RSMGWO	$4.9353\times 10^{-07}$	$1.1183\times 10^{-05}$	$4.9597\times 10^{-05}$	$1.0735\times 10^{-05}$
F13	MSIAR-GWO	$9.7146\times 10^{-18}$	$7.8304\times 10^{-16}$	$2.2089\times 10^{-15}$	$7.0182\times 10^{-16}$
	GWO	$2.8330\times 10^{-01}$	$6.6325\times 10^{-01}$	$9.3964\times 10^{-01}$	$1.8708\times 10^{-01}$
	ARO	$2.0653\times 10^{-04}$	$3.5166\times 10^{-03}$	$1.3719\times 10^{-02}$	$5.1884\times 10^{-03}$
	DBO	$1.6868\times 10^{-03}$	$7.9490\times 10^{-01}$	$1.7784\times 10^{00}$	$5.0244\times 10^{-01}$
	WOA	$2.0306\times 10^{-01}$	$5.8140\times 10^{-01}$	$1.3248\times 10^{00}$	$3.0981\times 10^{-01}$
	IGWO	$5.9025\times 10^{-05}$	$1.2100\times 10^{-01}$	$5.0931\times 10^{-01}$	$1.4167\times 10^{-01}$
	AGWO	$1.8477\times 10^{00}$	$2.1596\times 10^{00}$	$2.4269\times 10^{00}$	$1.5906\times 10^{-01}$
	RSMGWO	$5.8975\times 10^{-06}$	$9.6842\times 10^{-05}$	$4.3666\times 10^{-04}$	$1.1118\times 10^{-04}$

**Table 6 sensors-25-00892-t006:** Experimental comparison results of MSIAR-GWO and other advanced algorithms on fixed dimensional multimodal benchmark test functions.

Function	Algorithm	Best	Mean	Worst	St. dev
F14	MSIAR-GWO	$3.0749\times 10^{-04}$	$3.1346\times 10^{-04}$	$4.2433\times 10^{-04}$	$2.6103\times 10^{-05}$
	GWO	$3.0752\times 10^{-04}$	$3.4523\times 10^{-03}$	$2.0861\times 10^{-02}$	$7.3705\times 10^{-03}$
	ARO	$3.0749\times 10^{-04}$	$5.1870\times 10^{-04}$	$1.6051\times 10^{-03}$	$4.1789\times 10^{-04}$
	DBO	$3.0749\times 10^{-04}$	$6.9375\times 10^{-04}$	$2.2421\times 10^{-03}$	$4.5315\times 10^{-04}$
	WOA	$3.0865\times 10^{-04}$	$7.7236\times 10^{-04}$	$2.2368\times 10^{-03}$	$5.5450\times 10^{-04}$
	IGWO	$3.0749\times 10^{-04}$	$3.0749\times 10^{-04}$	$3.0752\times 10^{-04}$	$7.8376\times 10^{-09}$
	AGWO	$3.0830\times 10^{-04}$	$1.5350\times 10^{-03}$	$2.0942\times 10^{-02}$	$4.5939\times 10^{-03}$
	RSMGWO	$3.0775\times 10^{-04}$	$6.1889\times 10^{-04}$	$1.2235\times 10^{-03}$	$2.5088\times 10^{-04}$
F15	MSIAR-GWO	$3.9789\times 10^{-01}$	$3.9789\times 10^{-01}$	$3.9789\times 10^{-01}$	0
	GWO	$3.9789\times 10^{-01}$	$3.9789\times 10^{-01}$	$3.9789\times 10^{-01}$	$1.6683\times 10^{-06}$
	ARO	$3.9789\times 10^{-01}$	$3.9789\times 10^{-01}$	$3.9789\times 10^{-01}$	0
	DBO	$3.9789\times 10^{-01}$	$3.9789\times 10^{-01}$	$3.9789\times 10^{-01}$	0
	WOA	$3.9789\times 10^{-01}$	$3.9790\times 10^{-01}$	$3.9794\times 10^{-01}$	$1.6394\times 10^{-05}$
	IGWO	$3.9789\times 10^{-01}$	$3.9789\times 10^{-01}$	$3.9789\times 10^{-01}$	0
	AGWO	$3.9789\times 10^{-01}$	$3.9795\times 10^{-01}$	$3.9887\times 10^{-01}$	$2.1732\times 10^{-04}$
	RSMGWO	$3.9789\times 10^{-01}$	$3.9789\times 10^{-01}$	$3.9790\times 10^{-01}$	$3.1005\times 10^{-06}$
F16	MSIAR-GWO	$3.0000\times 10^{00}$	$3.0000\times 10^{00}$	$3.0000\times 10^{00}$	$1.4043\times 10^{-15}$
	GWO	$3.0000\times 10^{00}$	$3.0000\times 10^{00}$	$3.0001\times 10^{00}$	$3.3618\times 10^{-05}$
	ARO	$3.0000\times 10^{00}$	$3.0000\times 10^{00}$	$3.0000\times 10^{00}$	$3.6734\times 10^{-16}$
	DBO	$3.0000\times 10^{00}$	$4.3500\times 10^{00}$	$30.0000\times 10^{00}$	$6.0374\times 10^{00}$
	WOA	$3.0000\times 10^{00}$	$5.7011\times 10^{00}$	$30.0196\times 10^{00}$	$8.3138\times 10^{00}$
	IGWO	$3.0000\times 10^{00}$	$3.0000\times 10^{00}$	$3.0000\times 10^{00}$	$1.6710\times 10^{-15}$
	AGWO	$3.0000\times 10^{00}$	$4.3500\times 10^{00}$	$30.0003\times 10^{00}$	$6.0375\times 10^{00}$
	RSMGWO	$3.0000\times 10^{00}$	$3.0000\times 10^{00}$	$3.0000\times 10^{00}$	$2.5927\times 10^{-06}$
F17	MSIAR-GWO	−$3.3220\times 10^{00}$	−$3.3042\times 10^{00}$	−$3.2031\times 10^{00}$	$4.3556\times 10^{-02}$
	GWO	−$3.3220\times 10^{00}$	−$3.2742\times 10^{00}$	−$3.2015\times 10^{00}$	$6.0047\times 10^{-02}$
	ARO	−$3.3220\times 10^{00}$	−$3.2328\times 10^{00}$	−$3.2031\times 10^{00}$	$5.2820\times 10^{-02}$
	DBO	−$3.3220\times 10^{00}$	−$3.2086\times 10^{00}$	−$2.2671\times 10^{00}$	$2.3566\times 10^{-01}$
	WOA	−$3.3217\times 10^{00}$	−$3.2421\times 10^{00}$	−$2.8505\times 10^{00}$	$1.2519\times 10^{-01}$
	IGWO	−$3.3220\times 10^{00}$	−$3.3090\times 10^{00}$	−$3.2031\times 10^{00}$	$3.6545\times 10^{-02}$
	AGWO	−$3.3219\times 10^{00}$	−$3.2523\times 10^{00}$	−$3.1214\times 10^{00}$	$8.0853\times 10^{-02}$
	RSMGWO	−$3.3220\times 10^{00}$	−$3.2859\times 10^{00}$	−$3.2011\times 10^{00}$	$5.6460\times 10^{-02}$
F18	MSIAR-GWO	−$10.5364\times 10^{00}$	−$10.5364\times 10^{00}$	−$10.5364\times 10^{00}$	$1.4117\times 10^{-15}$
	GWO	−$10.5363\times 10^{00}$	−$10.5358\times 10^{00}$	−$10.5341\times 10^{00}$	$5.1591\times 10^{-04}$
	ARO	−$10.5364\times 10^{00}$	−$10.2660\times 10^{00}$	−$5.1285\times 10^{00}$	$1.2092\times 10^{00}$
	DBO	−$10.5364\times 10^{00}$	−$6.6572\times 10^{00}$	−$2.4217\times 10^{00}$	$3.0401\times 10^{00}$
	WOA	−$10.5341\times 10^{00}$	−$6.8041\times 10^{00}$	−$1.6633\times 10^{00}$	$3.1737\times 10^{00}$
	IGWO	−$10.5364\times 10^{00}$	−$10.5364\times 10^{00}$	−$10.5364\times 10^{00}$	$9.6189\times 10^{-10}$
	AGWO	−$10.5332\times 10^{00}$	−$8.9130\times 10^{00}$	−$2.4208\times 10^{00}$	$3.2891\times 10^{00}$
	RSMGWO	−$10.5364\times 10^{00}$	−$10.5354\times 10^{00}$	−$10.5311\times 10^{00}$	$1.2553\times 10^{-03}$

**Table 7 sensors-25-00892-t007:** Wilcoxon rank sum test *p*-values and performance between MSIAR-GWO and selected meta-heuristic algorithms.

Function	GWO	ARO	DBO	WOA	IGWO	AGWO	RSMGWO
	P/R	P/R	P/R	P/R	P/R	P/R	P/R
F1	$8.007\times 10^{-09}$	$8.007\times 10^{-09}$	$8.007\times 10^{-09}$	$8.007\times 10^{-09}$	$8.007\times 10^{-09}$	$8.007\times 10^{-09}$	NAN
	/+	/+	/+	/+	/+	/+	/=
F2	$8.007\times 10^{-09}$	$8.007\times 10^{-09}$	$8.007\times 10^{-09}$	$8.007\times 10^{-09}$	$8.007\times 10^{-09}$	$8.007\times 10^{-09}$	NAN
	/+	/+	/+	/+	/+	/+	/=
F3	$8.007\times 10^{-09}$	$8.007\times 10^{-09}$	$8.007\times 10^{-09}$	$8.007\times 10^{-09}$	$8.007\times 10^{-09}$	$8.007\times 10^{-09}$	NAN
	/+	/+	/+	/+	/+	/+	/=
F4	$8.007\times 10^{-09}$	$8.007\times 10^{-09}$	$8.007\times 10^{-09}$	$8.007\times 10^{-09}$	$8.007\times 10^{-09}$	$8.007\times 10^{-09}$	NAN
	/+	/+	/+	/+	/+	/+	/=
F5	$6.796\times 10^{-08}$	$6.796\times 10^{-08}$	$6.796\times 10^{-08}$	$6.796\times 10^{-08}$	$6.796\times 10^{-08}$	$6.796\times 10^{-08}$	$6.796\times 10^{-08}$
	/+	/+	/+	/+	/+	/+	/+
F6	$6.796\times 10^{-08}$	$6.796\times 10^{-08}$	$6.796\times 10^{-08}$	$6.796\times 10^{-08}$	$6.796\times 10^{-08}$	$6.796\times 10^{-08}$	$6.796\times 10^{-08}$
	/+	/+	/+	/+	/+	/+	/+
F7	$6.796\times 10^{-08}$	$1.576\times 10^{-06}$	$1.918\times 10^{-07}$	$1.657\times 10^{-07}$	$6.796\times 10^{-08}$	$1.116\times 10^{-03}$	$1.251\times 10^{-05}$
	/+	/+	/+	/+	/+	/+	/+
F8	$6.796\times 10^{-08}$	$2.745\times 10^{-04}$	$1.227\times 10^{-03}$	$5.979\times 10^{-01}$	$5.255\times 10^{-05}$	$6.796\times 10^{-08}$	$6.796\times 10^{-08}$
	/+	/+	/+	/−	/+	/+	/−
F9	$7.948\times 10^{-09}$	NAN	NAN	NAN	$8.007\times 10^{-09}$	NAN	NAN
	/+	/=	/=	/=	/+	/=	/=
F10	$7.746\times 10^{-09}$	NAN	NAN	$2.285\times 10^{-05}$	$6.702\times 10^{-09}$	$7.428\times 10^{-10}$	$2.412\times 10^{-08}$
	/+	/=	/=	/+	/+	/+	/+
F11	$4.532\times 10^{-03}$	NAN	NAN	NAN	$4.016\times 10^{-02}$	NAN	NAN
	/+	/=	/=	/=	/+	/=	/=
F12	$6.796\times 10^{-08}$	$6.796\times 10^{-08}$	$6.796\times 10^{-08}$	$6.796\times 10^{-08}$	$6.796\times 10^{-08}$	$6.796\times 10^{-08}$	$6.796\times 10^{-08}$
	/+	/+	/+	/+	/+	/+	/+
F13	$6.796\times 10^{-08}$	$6.796\times 10^{-08}$	$6.796\times 10^{-08}$	$6.796\times 10^{-08}$	$6.796\times 10^{-08}$	$6.796\times 10^{-08}$	$6.796\times 10^{-08}$
	/+	/+	/+	/+	/+	/+	/+
F14	$1.201\times 10^{-06}$	$3.066\times 10^{-06}$	$3.939\times 10^{-07}$	$1.918\times 10^{-07}$	$1.600\times 10^{-05}$	$7.948\times 10^{-07}$	$2.218\times 10^{-07}$
	/+	/+	/+	/+	/−	/+	/+
F15	$8.007\times 10^{-09}$	NAN	NAN	$8.007\times 10^{-09}$	NAN	$8.007\times 10^{-09}$	$8.007\times 10^{-09}$
	/+	/=	/=	/+	/=	/+	/+
F16	$5.629\times 10^{-08}$	$1.355\times 10^{-03}$	$2.672\times 10^{-02}$	$5.629\times 10^{-08}$	$2.240\times 10^{-01}$	$5.629\times 10^{-08}$	$5.629\times 10^{-08}$
	/+	/+	/+	/+	/−	8/+	/+
F17	$6.152\times 10^{-06}$	$1.508\times 10^{-04}$	$3.255\times 10^{-02}$	$9.074\times 10^{-06}$	$1.146\times 10^{-04}$	$4.144\times 10^{-06}$	$1.330\times 10^{-05}$
	/+	/+	/+	/+	/−	/+	/+
F18	$3.959\times 10^{-08}$	$1.843\times 10^{-02}$	$8.275\times 10^{-06}$	$3.959\times 10^{-08}$	$3.959\times 10^{-08}$	$3.959\times 10^{-08}$	$3.959\times 10^{-08}$
	/+	/+	/+	/+	/+	/+	/+
+/=/−	18/0/0	14/4/0	14/4/0	15/2/1	14/1/3	16/2/0	11/6/1

**Table 8 sensors-25-00892-t008:** Comparison experiment with traditional A* algorithm in grid environment.

	Algorithm	MSIAR-GWO	A*
Map Dimensions			
20 × 20		29.1038	30.3848
40 × 40		57.7521	59.2548

**Table 9 sensors-25-00892-t009:** Comparison experimental results in 20 × 20 raster environment.

Map Dimensions	Algorithm	The Optimal Length	Average Length of Path	Path Standard Deviation
20 × 20	MSIAR-GWO	28.0192	28.5610	0.5917
	GWO	28.0285	29.8550	1.4466
	ARO	28.0192	28.7696	0.7939
	DBO	28.0285	29.8079	1.4803
	WOA	29.3782	32.2861	1.9639
	IGWO	28.0285	28.7271	0.5717
	AGWO	28.0285	30.5612	2.0611
	RSMGWO	28.0285	29.6183	0.9251

**Table 10 sensors-25-00892-t010:** Comparative experimental results in 30 × 30 raster environment.

Map Dimensions	Algorithm	The Optimal Length	Average Length of Path	Path Standard Deviation
30 × 30	MSIAR-GWO	42.4762	44.0835	2.1519
	GWO	44.0427	53.6080	5.0156
	ARO	42.4762	46.0486	3.3236
	DBO	43.7280	50.0885	3.8749
	WOA	44.2012	51.5709	3.9140
	IGWO	43.0913	46.9421	2.9170
	AGWO	44.5422	51.8773	4.9659
	RSMGWO	43.7881	51.6112	4.5937

**Table 11 sensors-25-00892-t011:** Comparative experimental results in 40 × 40 raster environment.

Map Dimensions	Algorithm	The Optimal Length	Average Length of Path	Path Standard Deviation
40 × 40	MSIAR-GWO	60.0792	68.6196	4.0058
	GWO	70.0352	88.9720	11.7363
	ARO	60.1733	71.8339	6.2240
	DBO	66.8810	73.8673	2.5526
	WOA	67.6425	74.8505	2.7714
	IGWO	67.9427	85.1341	9.8011
	AGWO	72.6565	88.7081	10.6476
	RSMGWO	69.0529	87.9394	7.4126

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
