# Peer review of "Research on Mobile Robot Path Planning Based on MSIAR-GWO Algorithm"

_sensors, 2025, doi:10.3390/s25030892_

Round 1

Reviewer 1 Report

Comments and Suggestions for Authors

Contributions:

This manuscript proposes a reinforcement learning based multi-strategy improved grey wolf optimization algorithm (MSIAR-GWO) for mobile robot path planning. The main innovations include: 1) selecting a novel nonlinear convergence factor with adjustable parameters through reinforcement learning to balance exploration and exploitation; 2) Adaptive position update strategy based on detour foraging and dynamic weights, increasing the influence of better individuals and accelerating convergence; 3) The elimination and relocation strategy based on dynamic reverse learning of random gravity maintains population diversity and avoids falling into local optima. The experimental results show that the MSIAR-GWO algorithm outperforms other swarm intelligence optimization algorithms in benchmark testing and path planning experiments, demonstrating excellent stability, solution accuracy, and convergence speed.

Specific comments:

The paper addresses a practical problem and the organization is clear. Anyhow, the reviewer has some questions or suggestions about the manuscript:

1.      This manuscript does not describe the modeling of path planning problems and how to solve them using the proposed method. It is recommended that the author add relevant content.

2.      In the path planning experiment, the three designed test scenarios are relatively simple, and it seems that traditional graph search algorithms can exhibit better performance. It is recommended that the author improve the discrimination of the test scenarios.

3.      Some table data in the experimental results are suggested to be presented in the form of histograms for a more intuitive representation.

4.      The images in the article are blurry. It is recommended that the author replace them with clearer images.

Comments on the Quality of English Language

Many sentences in the manuscript are too lengthy and difficult for readers to understand. It is recommended that the author replace them with more concise expressions.

Author Response

Comment 1: This manuscript does not describe the modeling of path planning problems and how to solve them using the proposed method. It is recommended that the author add relevant content.

 Response: Thank you very much for your constructive advice. We have supplemented the text with relevant content describing the modeling of path planning problems and how to solve these problems using the proposed method. For the modeling process, it first requires the generation of a random node between each column of the map between the starting and ending points, and then all nodes are placed in order on the path as gray wolf individuals. During this process, each grey wolf in the population is represented by a -dimensional vector, such as the ith grey wolf individual is represented by, where  is the number of columns in the grid map. Secondly, after the generation of individuals, the path is made continuous and obstacle-free by interpolation. Moreover, the path is optimized by direct connection test method to further reduce redundancy, eliminate unnecessary inflection points and bends, and to obtain the shortest robot motion path, which makes it more efficient and concise, and thus improves the work efficiency of the mobile robot.

To further elaborate of the modeling process, the execution steps of the mobile robot path planning algorithm based on MSIAR-GWO are added as follows:

1.Set the parameters of the map, such as the size, starting position and

ending position, and the parameters of the MSIAR-GWO algorithm, such as the population size and the maximum number of iterations according to the map environment.

2.Calculate the value of each gray wolf individual according to Equation (32) and select the leader gray wolf based on the fitness. Determine the initial shortest path and the shortest path information. 

  1. Update the values of parameters a, A, and C according to Equations (16), (3) and (4), respectively.
  2. Update the position of each grey Wolf individual through Equation (20). If a<1, the derouting foraging strategy of Equation (24) and the greedy mechanism of Equation (28) are added to update the position of the current individual.
  3. The worst individuals are eliminated and re-updated by using the random centroid dynamic reverse learning based onEquation (30). 
  4. Calculate the path fitness function value to update the leader grey Wolf, and update the path shortest length and path shortest planning information.
  5. Determine whether the iteration termination condition is satisfied, if so, output the global shortest path length and path shortest planning information, otherwise, return to step 3 to continue optimization.
  6. The algorithm ends, and the best path planning result is output.

The above content can be seen in the blue-marked part of Section 4.3 of the modified draft.

Comment 2: In the path planning experiment, the three designed test scenarios are relatively simple, and it seems that traditional graph search algorithms can exhibit better performance. It is recommended that the author improve the discrimination of the test scenarios.

Response:

Thank you very much for your valuable comments. The size of the search space is mainly determined by the size of the map in the article, and the computational complexity of path planning increases with the number grids. Moreover, the number and distribution of obstacles in the environment have a direct impact on the difficulty of path planning. The complexity of the map is distinguished by setting different dimensions of 20×20, 30×30 and 40×40 in the draft, and the comparative experiment considers the comparison of grey wolf optimization algorithm and its related variants and other advanced intelligent optimization algorithms, but lacks the comparative with traditional graph search algorithms. According to the your opinions, the comparative experiment in the test scene of the algorithm proposed in this article and the traditional graph search algorithm-A* algorithm is added to the modified manuscript.

The comparison results given in the Table 8 and Figure 11 show that the traditional A* algorithm has limited efficiency and global optimization capability when facing complex environments in the test scenarios such as dense obstacles and multiple path choices. The MSIAR-GWO algorithm has competitiveness in solving the shortest path problem, especially showing stronger global optimization capability in complex environments.  

 The above contents can be find in the blue-marked part of Section 4.3.2 of the modified draft.

Comment 3: Some table data in the experimental results are suggested to be presented in the form of histograms for a more intuitive representation.

 Response: Thank you for your suggestions. To present the experimental results more intuitively, the paper supplements the histograms of the comparative experimental results in different grid environments, as well as the histograms of the performance comparison between MSIAR-GWO and other algorithms, etc. The above modifications have greatly improved the visual effect of the algorithm given in this work, making the effectiveness of the algorithm more convincing. Thank you again for your constructive suggestions.

The corresponding modifications can be seen from the Figure13-Figure 17 in the modified draft.

Comment 4: The images in the article are blurry. It is recommended that the author replace them with clearer images.

 Response: Thank you very much for your advice, and we apologize for our careless. In resubmitted manuscript, we have revised the manuscript and carefully checked the entire manuscript multiple times.

Reviewer 2 Report

Comments and Suggestions for Authors

This paper proposes a Multi-Strategy Improved Gray Wolf Optimization algorithm (MSIAR-GWO) based on reinforcement learning. Through innovative strategies such as intelligent parameter configuration, adaptive position update, Brownian motion, and Lévy flight perturbation mechanisms, the algorithm's global optimization capability and convergence precision in path planning are enhanced. Experimental results show that MSIAR-GWO outperforms traditional swarm intelligence optimization algorithms in terms of solution accuracy, convergence speed, and stability. For this paper, the following suggestions are provided:

1.    The images in the article are unclear, such as the text in Figure 5, which is difficult to read. Please review and address similar issues throughout the paper.

2.    The paper lacks visual explanation of path search and exploration behavior. It is recommended to add relevant visualizations to better illustrate the algorithm's performance.

3.    The paper lacks real-world physical experiments, and it is recommended to include them to enhance the persuasiveness of the study.

4.    The simulation experiments in Figure 13 compare different algorithms in the same scenario. Would it be possible to include additional experiments that compare the proposed algorithm in different scenarios to further validate its performance?

5.    Multiple random mechanisms are incorporated into the algorithm (such as Brownian motion, Lévy flight perturbation, and stochastic center of gravity dynamic reverse learning). It is recommended to add an analysis of the impact of these random factors on the algorithm's performance to ensure the stability and consistency of the algorithm under different operating conditions.

Author Response

Comment 1: The images in the article are unclear, such as the text in Figure 5, which is difficult to read. Please review and address similar issues throughout the paper.

Response: Thank you very much for your advice, and we apologize for our careless. In resubmitted manuscript, we have revised the manuscript and carefully checked the entire manuscript multiple times. 

 Comment 2: The paper lacks visual explanation of path search and exploration behavior. It is recommended to add relevant visualizations to better illustrate the algorithm's performance.

Response: We sincerely appreciate your feedback. Thank you very much for your constructive advice. We have supplemented the text with relevant content describing the modeling of path planning problems and how to solve these problems using the proposed method. For the modeling process, it first requires the generation of a random node between each column of the map between the starting and ending points, and then all nodes are placed in order on the path as gray wolf individuals. During this process, each grey wolf in the population is represented by a -dimensional vector, such as the ith grey wolf individual is represented as, where  is the number of columns in the grid map. Secondly, after the generation of individuals, the path is made continuous and obstacle-free by interpolation. Moreover, the path is optimized by direct connection test method to further reduce redundancy, eliminate unnecessary inflection points and bends, and to obtain the shortest robot motion path, which makes it more efficient and concise, and thus improves the work efficiency of the mobile robot.

To further elaborate of the modeling process, the execution steps of the mobile robot path planning algorithm based on MSIAR-GWO are added as follows:

  1. Parameter setting: set theparameters of the map, such as the size, starting position and

ending position, and the parameters of the MSIAR-GWO algorithm, such as the population size and the maximum number of iterations according to the map environment.

  1. Initialization ofthe population: calculate the value of each gray wolf individual according to Equation (32) and select the leader gray wolf based on the fitness. Determine the initial shortest path and the shortest path information. 
  2. 3.Update the values of parameters a, A, and C according to Equations(16), (3) and (4).
  3. Update the position of each grey wolf individual throughEquation (20). If a < 1, add the detour foraging strategy of formula (24) and the greedy mechanism according to formula (28) to update the position of the current individual.
  4. The worst individuals are eliminated and re-updated by using the random centroid dynamic reverse learning based onEquation (30). 
  5. Calculate the path fitness function value, update the leader grey wolf, and update the shortest path length and the shortest path planning information.
  6. Judge whether the iteration termination condition is met. If so, output the global shortest path length and the shortest path planning information, if not,return to step 3 to continue the optimization.
  7. The algorithm ends, and the best path planning result is output.

The above content can be seen in the blue-marked part of Section 4.3 of the modified draft.

Comment 3: The paper lacks real-world physical experiments, and it is recommended to include them to enhance the persuasiveness of the study.

 Response: Thank you for your meticulous review and valuable suggestions of our research. Regarding the suggestion of incorporating real-world physical experiments to enhance the persuasiveness the research, we fully agree with the value of this direction. Real environments often contain many other uncertain factors, such as dynamic obstacles and unknown static obstacles of the environment, which are not sufficient to be addressed only by global path planning algorithms. It is necessary to combine local path planning algorithms for autonomous obstacle avoidance to achieve true sense of autonomous navigation for mobile robots. Currently, our research group is carrying out research on local path planning algorithms and is also building the corresponding physical robot platform. Due to the limitations of the current experimental conditions and the complexity of building real environments, we are unable to supplement the experiments in real scenarios in this paper in the short term.

Despite this, in order to verify the practical application effect of the algorithm, we designed a variety of simulation experiments, which are based on highly abstracted environmental models and fully consider the key factors such as the dimension of the environment and the dense distribution of obstacles in real scenarios. Compared with traditional algorithms, the results show that our algorithm has significant advantages in terms of global search capability, computational efficiency, and path smooth in path planning. Therefore, we believe that the current experimental results can fully demonstrate the feasibility and effectiveness of the research, and provide reliable theoretical and experimental basis for further.

In future work, we will optimize the experimental conditions and further carry out physical experiments in real environments to more comprehensively verify the practical application value of the algorithm. Thank you again for your suggestions!

Comment 4: The simulation experiments in Figure 13 compare different algorithms in the same scenario. Would it be possible to include additional experiments that compare the proposed algorithm in different scenarios to further validate its performance?

Response: Thank you for your constructive suggestion. To evaluate the optimization performance of the MSIAR-GWO algorithm proposed in this work, it is compared with other common swarm optimization algorithms, such as the original grey wolf optimizer (GWO), artificial rabbit optimization (ARO), dung beetle optimization (DBO), and whale optimization algorithm (A), as well as with a series of other improved algorithms based on the grey wolf optimizer, such as IGWO, AGWO, and RSMGWO. 

To further verify the performance of the MSIAR-GWO algorithm, comparative experiments with traditional graph search algorithm-A* algorithm in the test scenario is supplemented in the revised version. Although the traditional A* algorithm has good path planning performance in simple scenarios, the comparative experimental results given in the Table 8 and Figure 11 show that the traditional A* algorithm has limited efficiency and global optimization capability when facing complex environments in the test scenarios such as dense obstacles and multiple path choices. The MSIAR-GWO algorithm has competitiveness in solving the shortest path problem, especially showing stronger global optimization capability in complex environments.

The above contents can be find in the blue-marked part of Section 4.3.2 of the modified draft.

Comment 5: Multiple random mechanisms are incorporated into the algorithm (such as Brownian motion, Lévy flight perturbation, and stochastic center of gravity dynamic reverse learning). It is recommended to add an analysis of the impact of these random factors on the algorithm's performance to ensure the stability and consistency of the algorithm under different operating conditions.

Response: Thank you for your constructive suggestion. Lévy flight is a random motion with a power-law distribution, where the step length follows a heavy-tailed distribution, occasionally taking larger jumps including occasional long-distance jumps, which can cover a larger range and has a high search efficiency. Brownian motion is a continuous and uniform random motion, where the step follows a normal distribution, the motion has no significant large jumps, each movement is relatively small, the diffusion is slower, and the coverage is relatively limited.

To visualize these two perturbations more transparently, the trajectory distribution map of

Lévy flight and Brownian motion with the same number of are supplemented, which can be seen from Figure 4 in the revised version. It can be clearly seen from the distribution map that under the condition of the same number of steps, the displacement of Lévy flight is much larger that of Brownian motion, and it can explore a larger space. Lévy flight can cover a larger area with fewer steps, which is very useful for exploring unknown.

This paper introduces the Levy flight or Brownian motion adaptively by the energy factor E, which makes the grey wolf individuals distribute widely in the search in the early stage of iteration, so as to improve the global optimization ability, and in the later stage of iteration, it makes the solution space more refined and continuous, which helps the algorithm to find better solutions in the local area and improve the accuracy of the algorithm.

The corresponding modification content can be find in the blue-marked part and Figure 4 in Section 3.2 from the revised draft.

Round 2

Reviewer 2 Report

Comments and Suggestions for Authors

NONE